# RoCA: Robust Cross-Domain End-to-End Autonomous Driving

**Rajeev Yasarla** [1]   **Shizhong Han** [1]   **Hsin-Pai Cheng** [1]   **Apratim Bhattacharyya** [1]   **Shweta Mahajan**† [1]   **Litian Liu** [1]
**Yunxiao Shi** [1]   **Risheek Garrepalli** [1]   **Hong Cai** [1]   **Fatih Porikli** [1]

## Abstract

End-to-end (E2E) autonomous driving has recently emerged as a new paradigm, offering significant potential. However, few studies have looked into the practical challenge of deployment across domains (e.g., cities). Although several works have incorporated Large Language Models (LLMs) to leverage their open-world knowledge, LLMs do not guarantee cross-domain driving performance and may incur prohibitive retraining costs during domain adaptation. In this paper, we propose RoCA, a novel framework for robust cross-domain E2E autonomous driving. RoCA formulates the joint probabilistic distribution over the tokens that encode ego and surrounding vehicle information in the E2E pipeline. Instantiating with a Gaussian process (GP), RoCA learns a set of basis tokens with corresponding trajectories, which span diverse driving scenarios. Then, given any driving scene, it is able to probabilistically infer the future trajectory. By using RoCA together with a base E2E model in source-domain training, we improve the generalizability of the base model, without requiring extra inference computation. In addition, RoCA enables robust adaptation on new target domains, significantly outperforming direct finetuning. We extensively evaluate RoCA on various cross-domain scenarios and show that it achieves strong domain generalization and adaptation performance.

## 1. Introduction

Moving beyond the traditional modular design, where distinct components like perception [Philion & Fidler, 2020; Yin et al., 2021; Li et al., 2022; Wang et al., 2025b], motion prediction [Chai et al., 2020; Liu et al., 2021; Ngiam et al., 2022], and planning [Sun et al., 2023] are often developed and optimized in isolation, the focus in autonomous driving research has recently shifted towards integrated, end-to-end (E2E) systems [Casas et al., 2021; Chitta et al., 2021; Chen & Krähenbühl, 2022; Hu et al., 2022; Wu et al., 2022; Jiang et al., 2023]. While E2E approaches can potentially provide enhanced overall driving performance thanks to the joint optimization across components, their robustness can be lacking when encountering less frequent scenarios. An important factor is the lack of diversity in existing large-scale training datasets *e.g.,* [Dosovitskiy et al., 2017; Caesar et al., 2020; Ettinger et al., 2021], which often fail to capture the full spectrum of driving scenarios. For instance, datasets like nuScenes [Caesar et al., 2020] are dominated by simple events, with limited coverage of rare, safety-critical edge cases. This imbalance is further amplified by standard training protocols, which tend to prioritize performance on frequent scenarios, causing the optimization to under-weigh long-tail events. As a result, E2E models trained in such a way have sub-optimal performance when deployed in different domains, such as different cities, lighting environments, camera characteristics, or weather conditions.

Large language models (LLMs) have recently emerged as a potentially powerful avenue to address these challenges, as their extensive world knowledge, gleaned from vast internet-scale data, may enable more effective generalization to unseen or rare scenarios [Wei et al., 2022]. Multi-modal variants (MLLMs) further enhance this by integrating visual inputs [Li et al., 2023; Liu et al., 2024], enabling a new wave of E2E driving systems that aim to achieve higher interpretability and improved long-tail robustness [Wang et al., 2023a; Sima et al., 2024; Tian et al., 2025; Hwang et al., 2024; Tian et al., 2024; Wang et al., 2025a; Hegde et al., 2025]. However, this integration introduces its own set of obstacles. One concern is that there is no inherent guarantee that these LLM-infused models will generalize across disparate domains without further adaptation. Moreover, retraining these massive models for new domains is often prohibitively expensive, demanding large quantities of specialized instruction-following data. Critically, even with broad world knowledge, the fundamental issue of data imbalance may persist, limiting model reliability in safety-

[1]Qualcomm AI Research, an initiative of Qualcomm Technologies, Inc. †Work was completed while employed at Qualcomm AI Research. Correspondence to: Rajeev Yasarla, Hong Cai <ryasarla@qti.qualcomm.com, hongcai@qti.qualcomm.com>.

*Proceedings of the 43$^{rd}$ International Conference on Machine Learning*, Seoul, South Korea. PMLR 306, 2026. Copyright 2026 by the author(s).

critical long-tail situations if not explicitly addressed during training.

To address these challenges, we propose RoCA (Robust Cross-domain end-to-end Autonomous driving). RoCA is an end-to-end autonomous driving framework designed for enhanced robustness and efficient adaptation using only multi-view images. Instead of relying on pre-trained LLMs, RoCA learns a compact yet comprehensive codebook of basis token embeddings ($\mathbf{b}$) that represent diverse ego states (pose, velocity, acceleration) and agent states (motion trajectory, velocity, acceleration), spanning both source and potentially target data characteristics. Crucially, RoCA leverages this learned codebook within a Gaussian Process (GP) framework. During inference, given a new scene's token embedding, the GP probabilistically predicts future ego waypoints and agent motion trajectories by leveraging the correlation between the current embedding and the learned basis embeddings ($\mathbf{b}$) and their associated known trajectories ($\mathbf{w} = g(\mathbf{b})$) for a learned mapping ($g(.)$) This probabilistic formulation inherently supports generalization, as predictions for novel scenes are informed by their similarity to known embeddings within the diverse codebook. Furthermore, the variance estimated by the GP provides a principled measure of prediction uncertainty. This variance can be used to dynamically weight the training loss, enabling RoCA to automatically assign greater importance to uncertain or difficult predictions, thereby effectively addressing the training imbalance towards common scenarios and improving performance on critical long-tail events.

The RoCA framework typically involves an initial stage to build the codebook and optimize GP parameters using source data, followed by efficient deployment or adaptation using only multi-view images processed through the learned GP component. This architecture also naturally lends itself to extensions for online streaming adaptation and active learning.

Our main contributions are summarized as follows:

- We propose RoCA, a novel framework for robust cross-domain end-to-end autonomous driving. Leveraging a Gaussian process (GP) formulation, RoCA captures the joint distribution over ego and agent tokens, which encode their respective future trajectories, enabling probabilistic prediction.

- By utilizing our GP to impose regularization on source-domain training, RoCA leads to more robust end-to-end planning performance both in domain and across domains.

- RoCA enables adaptation of the end-to-end model on a new target domain. Apart from standard finetuning, its uncertainty awareness makes it possible to select more

useful data in the active learning setup. Furthermore, RoCA also supports online adaptation.

- Through extensive evaluations (including both closed-loop and long-tail scenarios), RoCA consistently demonstrates robust performance across domains, for instance, transferring from simulation to real world, driving in different cities, and under challenging image degradations. Moreover, domain adaptation with RoCA is not only more effective yielding superior planning accuracy, but is also more efficient, as it leverages predictive uncertainty to prioritize the most informative data for fine-tuning.

## 2. Related Work

Autonomous driving (AD) systems have evolved from modular pipelines [Chai et al., 2020; Codevilla et al., 2019; Li et al., 2022] to end-to-end (E2E) architectures mapping sensor inputs to trajectories [Casas et al., 2021; Chitta et al., 2021]. Vision-only E2E models such as UniAD [Hu et al., 2023], VAD [Jiang et al., 2023], SSR [Li & Cui, 2025], and SparseDrive [Sun et al., 2025] improve efficiency but struggle with domain shifts and rare long-tail events [Ettinger et al., 2021]. Recent work addresses these challenges via graph-based transformers [Loh et al., 2024], domain-invariant objectives [Wang et al., 2023b; Huang et al., 2025], and normalization techniques [Ye et al., 2023; Qian et al., 2024], yet robust handling of safety-critical edge cases remains difficult. Gaussian Processes (GPs) provide principled uncertainty modeling [Williams & Rasmussen, 2006; Yasarla et al., 2020b; 2021; 2022a;b; 2024] and have aided domain adaptation in related fields [Kim et al., 2019; Ge & Sun, 2023; Liang et al., 2025; Yasarla & Patel, 2019; Yasarla et al., 2020a], but their integration into E2E AD is underexplored. This work introduces a GP-based module to enhance trajectory prediction and improve generalization to long-tail scenarios.

## 3. Proposed Approach: RoCA

We present RoCA, a novel, Gaussian process (GP)-based framework for cross-domain end-to-end autonomous driving. By using a set of basis tokens trained to span diverse driving scenarios, RoCA module probabilistically infers a trajectory for the current input scene. RoCA not only enhances the robustness of the trained E2E model, but also provides adaptation capability on a new domain.

### 3.1. Problem Formulation and Architecture

Consider a driving scene at time $t$ consisting of an ego vehicle and a set of surrounding agents. Given a sequence of multi-view images $I_t$, along with the ego status information (that includes pose, velocity, acceleration), the objective of

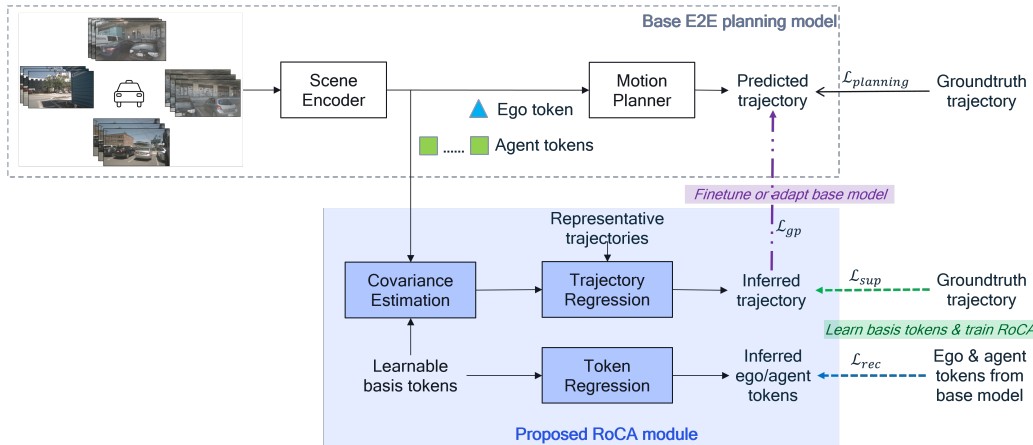

*Figure 1.* RoCA framework overview.[1] RoCA consists of two components. (1) A base E2E planner extracts the ego and agent tokens from multi-view images for the motion planner to predict future trajectories. (2) Proposed RoCA module, which leverages Gaussian process (GP). In source-domain training, RoCA learns a set of basis tokens from the source domain via reconstructing ego and agent tokens from the basis, supervised by the tokens from the base model (blue dashed arrow). Its GP-based trajectory regression model predicts trajectories which are supervised by ground-truth waypoints (green dashed arrow). During adaptation, RoCA generates pseudo ground truth to fine-tune the base model on the target domain (purple arrow).

RoCA is to jointly predict: (i) an ego trajectory $(p_w, c_w)$ that supports safe and efficient driving, and (ii) the motion trajectories of surrounding agents $(p_{w,a}, c_{w,a})$. Our proposed end-to-end (E2E) pipeline consists of two components: a base E2E model and the RoCA module. The base model, *e.g.,* [Jiang et al., 2023; Sun et al., 2025], typically includes: 1) a scene encoder $st(\cdot; \theta_{st})$, which converts input images into scene features/tokens, and 2) a motion planner $h(\cdot; \theta_h)$, which consumes these tokens to predict trajectories; $\theta_{st}$ and $\theta_h$ are learnable parameters. Figure 1 illustrates the overall system architecture.

**RoCA Preliminaries of GP.** To enhance robustness across domains, RoCA incorporates a Gaussian Process (GP) model. A GP can be viewed as an infinite collection of random variables such that any finite subset follows a joint Gaussian distribution. For a set of inputs $V = \{v_1, \ldots, v_n\}$, the corresponding function evaluations satisfy:

$$f(V) = [f(v_1), \ldots, f(v_n)]^\top \sim \mathcal{N}(\mu, K(V, V) + \sigma_\epsilon^2 I), \tag{1}$$

where $\mu$ denotes the mean vector and $K(V, V)$ is the covariance matrix induced by the kernel function. Predictions at unlabeled points are obtained by conditioning on the observed data, yielding a closed-form Gaussian posterior. The RoCA module instantiates this GP-based formulation as $g(\cdot; \theta_g, \kappa)$, where $\kappa(\cdot)$ is the kernel function and $\theta_g$ represents the learnable parameters.[1]

**Base E2E model.** The scene encoder extracts features from the input multi-camera images and cross-attends learnable queries/tokens with these features. Specifically, the scene encoder produces ego tokens e, and agent tokens a (among

---

[1]See [Seeger, 2004] for additional details on Gaussian Processes.

other possible tokens), which encode key information for the ego vehicle and of the other surrounding vehicles. The motion planner then takes the ego and agent tokens, and predicts the trajectories for both the ego and other vehicles: $p_{pred}, c_{pred}, p_{pred,a}, c_{pred,a} = h(e, a; \theta_h)$, where p denotes the waypoints and c denotes the trajectory class (*e.g.,* total number of classes can be 16 trajectory groups for each driving command of turn left, turn right, and go straight.)

**RoCA module.** The GP module contains learned basis tokens, as well as a set of candidate trajectories that one-to-one correspond to the basis tokens. The GP contains learned basis tokens, and for each of the token, there is a matching representative trajectory. By computing the correlation between e and a and the basis tokens via the kernel function $\kappa$, the GP conditionally infers the future ego and agent trajectories, $p_w, c_w, p_{w,a}, c_{w,a} = g_{ego}(e, a; \theta_g, \kappa)$.

Following [Sun et al., 2025], we use an anchor-based method to predict trajectories in both the base motion planner and RoCA module. More specifically, the model first classifies the future trajectory into one of the predefined groups: $N_{ego}$ groups for the ego car and $N_{agent}$ groups for other cars, and then, predicts a residual. The final predicted trajectory is obtained by adding the classified anchor trajectory and the predicted residual.

### 3.2. RoCA Module

In this part, we discuss our proposed RoCA module in details. This module allows to create an informed and diverse codebook of the plausible trajectories based on prior or pre-existing scenarios. The key advantage is that it can effectively infer trajectories under uncertainty or in new do-

mains by performing a similarity "lookup" to the basis in the codebook.

### 3.2.1. BASIS TOKENS AND TRAJECTORIES

We construct a "codebook" of learnable basis tokens, $\mathcal{B} = \{\mathbf{B}_k = \{\mathrm{b}_{j,k}\}_{j=1}^{C}\}_{k=1}^{N_{code}}$, where $N_{code}$ is the number of basis groups, each representing a certain trajectory pattern, *e.g.,* turn left, turn right, and $C$ is the group size. Each basis token is a $D$-dimensional vector $b_{j,k} \in \mathbb{R}^D$. These basis tokens should learn to span the space of ego and agent tokens, e and a, encountered across diverse driving scenes.

These basis tokens are designated to bijectively map to a set of plausible, safe driving trajectories, $\{\mathbf{W}_k\}_{k=1}^{N_{code}}$. To construct this set of basis trajectories, we first sample $N_{code} \cdot C$ representative trajectories from ground-truth train data, *e.g.,* nuScenes [Caesar et al., 2020]. They are then clustered into $N_{code}$ groups, such that each group $\mathbf{W}_k$ contains $C$ trajectories with similar driving patterns.

In our Gaussian process formulation, each trajectory $\mathrm{w}_{j,k} \in \mathbf{W}_k$ is associated with a unique, learnable basis $\mathrm{b}_{j,k} \in \mathbf{B}_k$. In other words, during training, each basis token learns the driving scenario that corresponds to its trajectory. With these basis tokens and trajectories, given a new driving scenario encoded by e and a, we can infer the ego and agent trajectories based on the correlation between the ego/agent tokens and the basis tokens. Within the $N_{code}$ groups, we designate $N_{ego}$ groups to represent distinct ego-car waypoint patterns and $N_{agent}$ groups for various types of agent trajectories.

### 3.2.2. RECONSTRUCTING EGO AND AGENT TOKENS

The basis tokens $\mathcal{B} = \{\mathbf{B}_k = \{\mathrm{b}_{j,k}\}_{j=1}^{C}\}_{k=1}^{N_{code}}$ should capture the manifold of ego and agent tokens across diverse driving scenarios. In order to train them, we derive the first set of losses via reconstruction of the original ego and agent tokens given by the base model from the basis tokens.

Given a driving scenario with ego and agent tokens from the base model, e and a, we first classify them into the respective basis groups. Specifically, the ego token is classified into one of the $N_{ego}$ groups and agent token into one of the $N_{agent}$ groups. Let $\mathrm{c_e}$ denote the index of the group assigned to e. This classification is performed based on the kernel distance metric and an MLP operating on distance, *i.e.,* $\mathrm{MLP}(\kappa(\mathrm{e}, \mathbf{B}))$ predicts the classification logits for the ego token (similarly for agent).

Let $\mathbf{B}_{\mathrm{c_e}}$ denote the basis tokens in the classified group $\mathrm{c_e}$. The core mechanism for learning the basis $\mathbf{B}_{\mathrm{c_e}}$ is by reconstructing the original ego token e using $\mathbf{B}_{\mathrm{c_e}}$ based on Gaussian process. The joint distribution of e and $\mathbf{B}_{\mathrm{c_e}}$ is

given by

$$p(\mathrm{e}, \mathbf{B}_{\mathrm{c_e}}) \sim \mathcal{N}\left(\begin{bmatrix} \mathrm{e} \\ \mathbf{B}_{\mathrm{c_e}} \end{bmatrix}, \begin{bmatrix} \kappa(\mathrm{e}) & \kappa(\mathrm{e}, \mathbf{B}_{\mathrm{c_e}}) \\ \kappa(\mathrm{e}, \mathbf{B}_{\mathrm{c_e}})^\top & \kappa(\mathbf{B}_{\mathrm{c_e}}) \end{bmatrix}\right), \tag{2}$$

where $p(.)$ denotes probability density function and $\kappa(.,.)$ is the kernel function evaluating pairwise distances among tokens (specifically, we use the RBF kernel).

The predictive mean $\hat{\mathrm{e}}$ (*i.e.,* the reconstruction of e) and predictive variance $\sigma_{\mathrm{e}}^2$ are given by

$$\hat{\mathrm{e}} = \mathbf{b}_{anchor,c_e} + \kappa(\mathrm{e}, \mathbf{B}_{\mathrm{c_e}})\kappa(\mathbf{B}_{\mathrm{c_e}})^{-1}\bar{\mathbf{B}}_{\mathrm{c_e}},$$
$$\sigma_e^2 = \kappa(\mathrm{e}) - \kappa(\mathrm{e}, \mathbf{B}_{\mathrm{c_e}})\kappa(\mathbf{B}_{\mathrm{c_e}})^{-1}\kappa(\mathrm{e}, \mathbf{B}_{\mathrm{c_e}})^\top + \sigma_{noise}^2\mathbb{I}, \tag{3}$$

where $\mathbf{b}_{anchor,c_e}$ is the mean of the tokens in group $c_e$, $\bar{\mathbf{B}}_{\mathrm{c_e}}$ is the zero-mean version of $\mathbf{B}_{\mathrm{c_e}}$, and $\sigma_{noise}^2$ is a small, learnable noise variance.

This prediction $\hat{\mathrm{e}}$ serves as an approximation of the original e, reconstructed with the basis tokens. We supervise this reconstruction with the original ego token. Similarly, we applying the same reconstruction process to obtain $\hat{\mathrm{a}}$ and $\sigma_{\mathrm{a}}$ for each agent token a, using their respective classified group of basis tokens $\mathbf{B}_{\mathrm{c_a}}$. The overall reconstruction loss for training the basis tokens is given by

$$\mathcal{L}_{rec} = \frac{1}{\sigma_{\mathrm{e}}^2}|\hat{\mathrm{e}} - \mathrm{e}|^2 + \log(\sigma_{\mathrm{e}}) + \frac{1}{\sigma_{\mathrm{a}}^2}|\hat{\mathrm{a}} - \mathrm{a}|^2 + \log(\sigma_{\mathrm{a}})$$
$$+ ||\mathbf{B}_{\mathrm{c_a}}\mathbf{B}_{\mathrm{c_a}}^\top - \mathbb{I}||^2 + ||\mathbf{B}_{\mathrm{c_e}}\mathbf{B}_{\mathrm{c_e}}^\top - \mathbb{I}||^2, \tag{4}$$

where the first four terms are based on maximum likelihood under Gaussian assumption and the last two terms encourages orthogonality of the basis. When learning the basis tokens and the parameters in the GP, we treat the original ego and agent tokens e and a as fixed targets, *i.e.,* no gradients flow through them.

### 3.2.3. TRAJECTORY PREDICTION VIA GAUSSIAN PROCESS

Similar to the previous part, given the ego and agent tokens from the base model, $e$ and $a$, we first classify them to their respective basis groups, $\mathrm{c_e}$ and $\mathrm{c_a}$. A GP-based regression then infers the future trajectory based on the correlation between the ego/agent token and the basis tokens. The predicted mean and variance for the ego trajectory, $\hat{\mathrm{p}}_{\mathrm{e}}$ and $\sigma_{\mathrm{e}}$, is given by

$$\hat{\mathrm{p}}_{\mathrm{w}} = \mathbf{w}_{anchor,c_e} + \kappa(\mathrm{e}, \mathbf{B}_{\mathrm{c_e}})\kappa(\mathbf{B}_{\mathrm{c_e}})^{-1}\bar{\mathbf{W}}_{\mathrm{c_e}},$$
$$\sigma_w^2 = \kappa(\mathrm{e}) - \kappa(\mathrm{e}, \mathbf{B}_{\mathrm{c_e}})\kappa(\mathbf{B}_{\mathrm{c_e}})^{-1}\kappa(\mathrm{e}, \mathbf{B}_{\mathrm{c_e}})^\top + \sigma_{noise}^2\mathbb{I}, \tag{5}$$

where $\mathbf{w}_{anchor,c_e}$ is the mean of the trajectories in group $\mathrm{c_e}$, $\bar{\mathbf{W}}_{\mathrm{c_e}}$ is the zero-mean version of $\mathbf{W}_{\mathrm{c_e}}$, and $\sigma_{noise}^2$ is a small, learnable noise variance. The predicted agent trajectory $\hat{\mathrm{p}}_{w,a}$ and variance $\sigma_{w,a}^2$ can be obtained similarly.

When training in the source domain, we supervise these GP-based trajectory predictions with the ground truth, as follows:

$$\mathcal{L}_{sup} = \frac{1}{\sigma_{\text{w}}^2}\mathcal{L}_{plan}(\hat{\text{p}}_{\text{w}}, \text{p}_{gt}) + \log(\sigma_{\text{w}}) + \mathcal{L}_{cls}(\text{c}_{\text{e}}, \text{c}_{gt,e})$$
$$+ \frac{1}{\sigma_{\text{w,a}}^2}\mathcal{L}_{mot}(\hat{\text{p}}_{\text{w,a}}, \text{p}_{gt,a}) + \log(\sigma_{\text{w,a}}) + \mathcal{L}_{cls}(\text{c}_{\text{a}}, \text{c}_{gt,a})$$
$$+ \mathcal{L}_{tpt}(\text{c}_{\text{e}}, \text{c}_{\text{p}}, \text{c}_{\text{n}}) + \mathcal{L}_{tpt}(\text{c}_{\text{a}}, \text{c}_{\text{p,a}}, \text{c}_{\text{n,a}})$$
$$(6)$$

where $\text{p}_{gt}$ and $\text{p}_{gt,a}$ are the ground-truth ego and agent trajectories, $\text{c}_{gt}$ and $\text{c}_{gt,a}$ are ground-truth ego and agent token categories. The predictive trajectory mean and variance are supervised using variance-weighted losses, similar to those used in [Jiang et al., 2023; Sun et al., 2025]). $\mathcal{L}_{plan}$ and $\mathcal{L}_{mot}$ denote the waypoint planning and motion tracking losses, as used in [Jiang et al., 2023; Sun et al., 2025]. To further refine the embedding space, we utilize triplet loss [Schroff et al., 2015]. For each ego/agent class prediction, we identify three positive classes ($\text{c}_{\text{p}}$), which exhibit similar driving patterns as the ground-truth class (*e.g.,* turn left with slightly different angles), and three negative classes ($\text{c}_{\text{n}}$), which have driving behaviors different from the true class (*e.g.,* turn left vs. turn right). The triplet loss encourages greater separation between distinct driving categories, and tighter clustering among the same class or similar classes.

### 3.3. Training and Adaptation

#### 3.3.1. TRAINING IN SOURCE DOMAIN

**Pre-training base E2E model.** First, we train the base E2E model on the source domain data following standard training procedure, *e.g.,* [Jiang et al., 2023; Li & Cui, 2025; Sun et al., 2025].

**Learning basis tokens and GP parameters.** Secondly, we use both $\mathcal{L}_{rec}$ of Eq. 4 and $\mathcal{L}_{sup}$ of Eq. 6 to train RoCA. This includes training the basis tokens and other parameters, *e.g.,* MLP parameters, kernel parameters.

**Finetuning base E2E model.** Finally, given the trained RoCA module, we utilize it to perform regularized finetuning. More specifically, in addition to the standard supervised loss used to train the base model, we additionally use the following loss by treating RoCA as a teacher:

$$\mathcal{L}_{gp} = \mathcal{L}_{cls}(\text{c}_{pred}, \text{c}_{\text{e}}) + \frac{1}{\sigma_{\text{w}}^2}\mathcal{L}_{plan}(\text{p}_{pred}, \hat{\text{p}}_{\text{w}}) + \log(\sigma_{\text{w}})$$
$$+ \mathcal{L}_{cls}(\text{c}_{pred,a}, \text{c}_{\text{a}}) + \frac{1}{\sigma_{\text{w,a}}^2}\mathcal{L}_{mot}(\text{p}_{pred,a}, \hat{\text{p}}_{\text{w,a}})$$
$$+ \log(\sigma_{\text{w,a}}) + \mathcal{L}_{tpt}(\text{c}_{pred}, \text{c}_{\text{p}}, \text{c}_{\text{n}}) + \mathcal{L}_{tpt}(\text{c}_{\text{a}}, \text{c}_{\text{p,a}}, \text{c}_{\text{n,a}})$$
$$+ D_{KL}(\text{c}_{pred,e}||\text{c}_{\text{e}}) + D_{KL}(\text{c}_{pred,a}||\text{c}_{\text{a}}), \quad (7)$$

where $\text{p}_{pred}$, $\text{c}_{pred}$, $\text{p}_{pred,a}$, and $\text{c}_{pred,a}$ are the predicted ego and agent trajectory waypoints and classes from the

base E2E model, $D_{KL}$ is the KL-divergence. This loss encourages the base model's prediction to align with the probabilistic prediction by the trained Gaussian process, which improves prediction robustness and regularizes against noise in training data.

#### 3.3.2. ADAPTATION IN TARGET DOMAIN

When ground-truth waypoints are available in the target domain, *e.g.,* based on ego status tracking. In such cases, model adaptation is then the same as the final step in source-domain training, where the standard ground-truth supervision on planning in Eq. 6 is used together with the GP-based regularization in Eq. 7. The final loss $\mathcal{L} = \mathcal{L}_{sup} + \mathcal{L}_{gp}$.

There are scenarios where ground-truth trajectories are not available. For instance, it is nontrivial to process large-volume driving logs and thus, ground-truth waypoints may not be available right after data is collected in the target domain (while images are usually readily available). As another example, in an online setting, the E2E driving model streams through video frames as the car drives and does not have access to the ground-truth waypoints on the fly. For unsupervised domain adaptation, as ground-truth labels are not available, we use $\mathcal{L}_{gp}$ to update the base E2E model.

## 4. Experiments

We conduct a thorough evaluation of RoCA in both closed-loop and open-loop planning settings. Our experiments demonstrate the domain adaptation capabilities of RoCA as a general plug-and-play framework that can be integrated with different end-to-end (E2E) autonomous driving models. Specifically, we present closed-loop evaluations on Bench2Drive, followed by cross-domain experiments that include simulation-to-real transfer from Bench2Drive to nuScenes, cross-city adaptation using nuScenes, and domain generalization under image degradations such as adverse weather, low-light conditions, and motion blur. In addition, we evaluate RoCA in an active learning setting for cross-domain adaptation, leveraging uncertainty estimates from our GP-based formulation.

### 4.1. Experimental Setup

**Datasets.** We use Bench2Drive [Jia et al., 2024] for closed-loop evaluation, which leverages the CARLA simulator and features 44 difficult interactive scenarios across diverse weather and urban conditions.[2] We utilize its official base training set (1,000 clips) for a fair comparison with other baselines, and evaluate performance on 220 challenging routes. To assess open-loop planning, we use the nuScenes dataset [Caesar et al., 2020], which consists of 28,000 samples with 22k/6k training/validation splits. nuScenes con-

---

[2]See Table 2 in Bench2Drive paper for more details

*Table 1.* Closed-loop evaluation on the challenging 220 routes of Bench2Drive [Jia et al., 2024]. "L" means large configuration for the corresponding model.

| Method | Closed-loop metric | | | Open-loop metric | Ability (%)↑ | | | | | |
|---|---|---|---|---|---|---|---|---|---|---|
| | DS↑ | SR ↑ | Efficiency↑ | Avg. L2↓ | Merging | Overtaking | Emergency Brake | Give Way | Traffic Sign | Mean |
| VAD [Jiang et al., 2023] | 42.35 | 15.0 | 157.94 | 0.91 | 8.11 | 24.44 | 18.64 | 20.00 | 19.15 | 18.07 |
| SSR [Li & Cui, 2025] | 47.71 | 24.4 | 97.41 | 0.86 | 19.10 | 29.38 | 41.67 | 30.00 | 56.75 | 35.38 |
| SparseDrive-S [Sun et al., 2025] | 51.01 | 27.8 | 103.1 | 0.83 | 22.64 | 30.38 | 44.56 | 30 | 53.81 | 36.28 |
| GenAD [Yang et al., 2024] | 44.81 | 15.9 | – | - | - | - | - | - | - | - |
| DriveTransformer-L [Jia et al., 2025] | 63.46 | 35.0 | 100.64 | 0.62 | 17.57 | 35.00 | 48.36 | 40.00 | 52.10 | 38.60 |
| ORION [Fu et al., 2025] | 77.74 | 54.6 | 151.48 | 0.68 | 25.00 | 71.11 | 78.33 | 30.00 | 69.15 | 54.72 |
| RoCA (VAD) | 56.90 | 34.3 | 175.42 | 0.74 | 27.39 | 43.91 | 53.76 | 30 | 47.17 | 40.45 |
| RoCA (SSR) | 59.81 | 41.0 | 110.61 | 0.69 | 31.96 | 48.23 | 60.94 | 40 | 63.75 | 48.98 |
| RoCA (ORION) | 80.38 | 58.2 | 181.06 | 0.57 | 34.06 | 74.91 | 83.76 | 40 | 72.83 | 61.11 |

*Table 2.* Sim-to-real model performance in zero-shot and fine-tuned settings from Bench2Drive [Jia et al., 2024] to nuScenes [Caesar et al., 2020]. For fine-tuned results, values in parentheses denote our domain adaptation performance without using ground-truth labels.

| Method | **Zero-shot** | | | | **Fine-tuned** | | | |
|---|---|---|---|---|---|---|---|---|
| | Full Val | | Targeted Val | | Full Val | | Targeted Val | |
| | Avg. L2 ↓ | Avg. Col. ↓ | Avg. L2 ↓ | Avg. Col. ↓ | Avg. L2 ↓ | Avg. Col. ↓ | Avg. L2 ↓ | Avg. Col. ↓ |
| VAD-Tiny [Jiang et al., 2023] | 1.32 | 0.51 | 1.59 | 0.54 | 0.91 | 0.39 | 1.27 | 0.39 |
| SSR [Li & Cui, 2025] | 1.08 | 0.31 | 1.47 | 0.44 | 0.75 | 0.15 | 1.19 | 0.37 |
| SparseDrive-S [Sun et al., 2025] | 1.17 | 0.34 | 1.51 | 0.49 | 0.65 | 0.14 | 0.85 | 0.31 |
| DiMA [Hegde et al., 2025] | 0.94 | 0.26 | 1.29 | 0.38 | 0.61 | 0.19 | 0.94 | 0.30 |
| DriveTransformer-S [Jia et al., 2025] | 1.12 | 0.33 | 1.44 | 0.43 | 0.69 | 0.20 | 0.95 | 0.33 |
| ORION [Fu et al., 2025] | 0.98 | 0.44 | 1.56 | 0.51 | 0.72 | 0.29 | 1.12 | 0.37 |
| RoCA (VAD-Tiny) | 0.85 | 0.24 | 1.19 | 0.34 | 0.63 (0.73) | 0.12 (0.17) | 0.88 (0.95) | 0.24 (0.27) |
| RoCA (SSR) | 0.79 | 0.22 | 1.10 | 0.32 | 0.54 (0.63) | 0.09 (0.13) | 0.65 (0.77) | 0.25 (0.29) |
| RoCA (SparseDrive-S) | 0.75 | 0.22 | 0.98 | 0.30 | 0.55 (0.64) | 0.10 (0.15) | 0.64 (0.80) | 0.24 (0.27) |
| RoCA (ORION) | 0.68 | 0.23 | 0.94 | 0.30 | 0.44 (0.52) | 0.08 (0.11) | 0.56 (0.72) | 0.20 (0.24) |

tains data collected from Boston and Singapore, which allows us to evaluate cross-domain generalization and adaptation across cities. Within the nuScenes validation set, we also consider a "targeted" subset containing 689 samples where the vehicle must make a turn, as established in [Weng et al., 2024]. To further evaluate robustness, we also use degraded versions of the validation set using controlled image corruptions such as adverse weather (like snow and fog), motion blur and low light from [Xie et al., 2025].

**Evaluation protocol.** For closed-loop evaluation metrics, we adopt the protocol proposed by [Jia et al., 2025], and evaluate the driving score (DS) and efficiency on Dev10. For open-loop evaluation metrics, we adopt the standardized evaluation proposed by [Weng et al., 2024] to compute two metrics, the L2 error (in meters) between the predicted and ground-truth waypoints, and the collision rate (in %) between the ego-vehicle and the surrounding vehicles. We compute open-loop metrics for comparisons on Bench2Drive and nuScenes validation.

**Model details.** We consider four recent and representative E2E planning models as our base architectures: ORION [Fu et al., 2025], SSR [Li & Cui, 2025], SparseDrive [Sun et al., 2025], and VAD [Jiang et al., 2023]. In our experiments, we adopt the *tiny* configuration of VAD (VAD-Tiny) and the *small* configuration of SparseDrive (SparseDrive-S). Note that our proposed RoCA can be used with any E2E planning model, as long as it provides a tokenized representation. When there are no explicit ego/agent tokens, we can apply RoCA over the queries or tokens that are fed into the planner. When we implement the codebook in RoCA, for

ego tokens, we use 16 groups for each of the driving commands: turn left, turn right, and go straight, resulting in $N_{ego} = 48$. We use $N_{agent} = 64$ groups to capture various types of agent trajectories. The total groups in the codebook is $N_{code} = N_{ego} + N_{agent} = 112$ and we set the group size $C = 64$. Each basis has dimension of $D = 256$.

### 4.2. Closed-Loop Evaluations on Bench2Drive

Table 1 shows the results on the challenging 220 routes of Bench2Drive. In this case, we only use RoCA as a training regularization and no adaptation is conducted during test time. We see that by using RoCA, we significantly improve the planning performance, e.g., improving driving score from 77.76 to 80.38 with ORION as the base model. This table highlights similar trends: RoCA (ORION) improves mean ability by 11.7% over the baseline ORION (i.e. $54.72 \rightarrow 61.11$), while RoCA (SSR) delivers a 27.8% (i.e. $35.38 \rightarrow 48.98$) improvement over SSR. Performance gains span critical skills such as merging, overtaking, emergency braking, yielding, and traffic-sign compliance—underscoring that GP module of RoCA predicts better posterior for ego waypoint trajectories.

### 4.3. Sim-to-Real Evaluation

We conduct a sim-to-real experiment by transferring models trained on Bench2Drive to the nuScenes dataset. The base E2E model, state-of-the-art baselines, and the proposed RoCA module are first trained on the source domain (Bench2Drive) for 12 epochs. In the target do-

*Table 3.* Cross-city planning performance on nuScenes dataset [Caesar et al., 2020].

| Method | Adapt | Boston → Singapore | | Singapore → Boston | |
|---|---|---|---|---|---|
| | | Avg. L2 (m)↓ | Avg. Col.(%)↓ | Avg. L2 (m)↓ | Avg. Col.(%)↓ |
| Senna [Jiang et al., 2024] | ✗ | 1.03 | 0.26 | 0.96 | 0.24 |
| w/ finetuning using GT | ✓ | 0.98 | 0.23 | 0.69 | 0.19 |
| DiMA [Hegde et al., 2025] | ✗ | 1.15 | 0.23 | 0.92 | 0.22 |
| w/ finetuning using GT | ✓ | 1.01 | 0.21 | 0.72 | 0.17 |
| VAD-Tiny [Jiang et al., 2023] | ✗ | 1.25 | 0.43 | 1.16 | 0.23 |
| w/ finetuning using GT | ✓ | 1.19 | 0.26 | 1.11 | 0.19 |
| w/ RoCA training regularization | ✗ | 1.17 | 0.21 | 1.01 | 0.19 |
| w/ RoCA unsupervised adaptation | ✓ | 1.02 | 0.19 | 0.94 | 0.17 |
| w/ RoCA adaptation using GT | ✓ | 0.94 | 0.17 | 0.89 | 0.16 |
| SparseDrive-S [Sun et al., 2025] | ✗ | 0.91 | 0.18 | 1.02 | 0.24 |
| w/ finetuning using GT | ✓ | 0.55 | 0.12 | 0.67 | 0.12 |
| w/ RoCA training regularization | ✗ | 0.79 | 0.15 | 0.88 | 0.15 |
| w/ RoCA unsupervised adaptation | ✓ | 0.71 | 0.13 | 0.68 | 0.11 |
| w/ RoCA adaptation using GT | ✓ | 0.49 | 0.09 | 0.51 | 0.10 |

*Table 4.* Cross-city active learning performance, using 5%, 10%, and 15% target training samples selected randomly or based on predictive variance by RoCA. The base model is SparseDrive-S in this case.

| Adapt. Method | Sampling | 5% | | 10% | | 15% | |
|---|---|---|---|---|---|---|---|
| | | Avg. L2 (m)↓ | Avg. Col.(%)↓ | Avg. L2 (m)↓ | Avg. Col.(%)↓ | Avg. L2 (m)↓ | Avg. Col.(%)↓ |
| **Singapore → Boston** | | | | | | | |
| Direct finetune | random | 0.767 | 0.215 | 0.753 | 0.199 | 0.711 | 0.175 |
| Direct finetune | RoCA | 0.745 | 0.191 | 0.719 | 0.183 | 0.678 | 0.126 |
| RoCA | random | 0.644 | 0.123 | 0.584 | 0.121 | 0.552 | 0.121 |
| RoCA | RoCA | 0.617 | 0.110 | 0.554 | 0.110 | 0.513 | 0.108 |
| **Boston → Singapore** | | | | | | | |
| Direct finetune | random | 0.891 | 0.198 | 0.839 | 0.201 | 0.823 | 0.185 |
| Direct finetune | RoCA | 0.828 | 0.192 | 0.815 | 0.172 | 0.793 | 0.166 |
| RoCA | random | 0.707 | 0.148 | 0.656 | 0.133 | 0.633 | 0.126 |
| RoCA | RoCA | 0.673 | 0.135 | 0.604 | 0.113 | 0.561 | 0.102 |

*Table 5.* Planning performance under image degradations including low light, motion blur, and under adverse weather conditions.

| Model | Adapt | lowlight | | motion blur | | Snow | | Fog | |
|---|---|---|---|---|---|---|---|---|---|
| | | Avg. L2 (m)↓ | Avg. Col.(%)↓ | Avg. L2 (m)↓ | Avg. Col.(%)↓ | Avg. L2 (m)↓ | Avg. Col.(%)↓ | Avg. L2 (m)↓ | Avg. Col.(%)↓ |
| **Full Val** | | | | | | | | | |
| SparseDrive-S | ✗ | 0.577 | 0.145 | 0.729 | 0.369 | 0.857 | 0.192 | 0.809 | 0.349 |
| w/ supervision using GT | ✓ | 0.547 | 0.228 | 0.573 | 0.118 | 0.610 | 0.111 | 0.704 | 0.272 |
| w/ RoCA training regularization | ✗ | 0.564 | 0.129 | 0.671 | 0.208 | 0.729 | 0.173 | 0.750 | 0.253 |
| w/ RoCA unsupervised adaptation | ✓ | 0.531 | 0.098 | 0.589 | 0.146 | 0.628 | 0.121 | 0.682 | 0.173 |
| w/ RoCA supervised adaptation using GT | ✓ | 0.526 | 0.090 | 0.541 | 0.104 | 0.581 | 0.096 | 0.619 | 0.151 |
| **Targeted Val** | | | | | | | | | |
| SparseDrive-S | ✗ | 0.712 | 0.325 | 0.832 | 0.389 | 0.907 | 0.287 | 0.939 | 0.399 |
| w/ supervision using GT | ✓ | 0.707 | 0.291 | 0.721 | 0.282 | 0.729 | 0.188 | 0.772 | 0.256 |
| w/ RoCA training regularization | ✗ | 0.703 | 0.228 | 0.766 | 0.248 | 0.849 | 0.208 | 0.853 | 0.220 |
| w/ RoCA unsupervised adaptation | ✓ | 0.626 | 0.188 | 0.733 | 0.203 | 0.756 | 0.135 | 0.721 | 0.166 |
| w/ RoCA supervised adaptation using GT | ✓ | 0.591 | 0.169 | 0.696 | 0.192 | 0.660 | 0.119 | 0.641 | 0.141 |

main (nuScenes), we evaluate both zero-shot performance (*i.e.,* no adaptation) and performance after 12 epochs of adaptation/fine-tuning. Specifically, we compare: (i) the baseline model, (ii) RoCA with source-only training regularization, and (iii) adaptation approaches with and without ground-truth labels. We also report results on the more challenging targeted split of nuScenes. Table 2 shows that RoCA consistently achieves superior sim-to-real performance compared to baselines and the LLM-based ORION model. For example, in zero-shot settings on the full validation split, RoCA improves average L2 error from 0.98 (ORION) to 0.68, and on the targeted split from 1.56 to 0.94. After fine-tuning, RoCA further reduces error to 0.44 (0.51 without ground truth) on full validation and 0.56 (0.72 without ground truth) on the targeted split, outperforming ORION

and other baselines. Moreover, even without ground-truth labels during adaptation, RoCA maintains strong performance (values in parentheses), demonstrating its robustness and effectiveness in unsupervised domain adaptation.

## 4.4. Closed-Loop Cross-Domain Evaluation on NAVSIM

To further evaluate performance under realistic interactive conditions, we conduct a closed-loop cross-domain evaluation by training on Bench2Drive (source) and evaluating on NAVSIM v1 [Dauner et al., 2024] (target). We adopt planning-oriented metrics—No-at-fault Collisions (NC↑), Ego Progress (EP↑), and PDMS↑—which better reflect closed-loop performance compared to displacement-based open-loop metrics. For RoCA-Orion and RoCA-

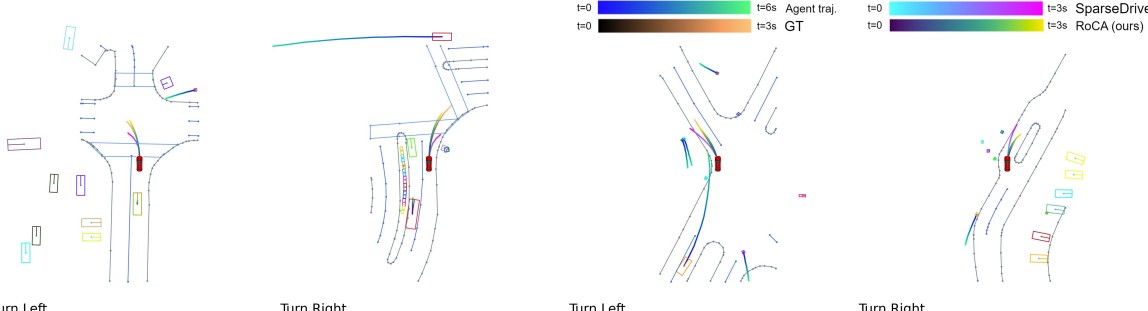

*Figure 2.* Visualization of sample planning results. Left (Right) two scenarios are in Boston (Singapore); note that t is right (left) driving in Boston (Singapore). The red car is the ego vehicle. The color gradient indicates the temporal horizon of the trajectory.

*Table 6.* Cross-domain closed-loop performance on NAVSIM v1 (Bench2Drive → NAVSIM). Higher is better.

| Method | NC↑ | EP↑ | PDMS↑ |
|---|---|---|---|
| SparseDrive | 97.1 | 74.2 | 82.1 |
| Orion | 97.9 | 78.7 | 84.4 |
| RoCA-SparseDrive | 98.0 | 77.5 | 85.8 |
| RoCA-Orion | **98.4** | **82.6** | **87.9** |

*Table 7.* Cross-domain closed-loop evaluation on DriveArena (Bench2Drive → DriveArena). Higher is better.

| Method | RC↑ | PDMS↑ |
|---|---|---|
| SparseDrive | 13.7 | 68.3 |
| Orion | 17.9 | 71.6 |
| RoCA-SparseDrive | 28.1 | 84.4 |
| RoCA-Orion | **33.7** | **91.2** |

SparseDrive, we additionally perform unsupervised fine-tuning following Sections 3.2.3 and 3.3.2. As shown in Table 6, RoCA consistently improves performance across all metrics. In particular, RoCA improves PDMS by +3.7 over SparseDrive (85.8 vs. 82.1) and by +3.5 over Orion (87.9 vs. 84.4), while also achieving gains in EP (+2.3 / +3.2) and NC (+0.9 / +0.5). These results demonstrate stronger closed-loop planning under domain shift, including robustness to differences in sensor configurations and interaction dynamics. Evaluating on NAVSIM provides a more realistic assessment beyond nuScenes, and we plan to extend this analysis to NAVSIM v2 in future work.

### 4.5. Closed-Loop Evaluation in DriveArena

To further evaluate performance under realistic interactive conditions, we conduct closed-loop cross-domain experiments using the DriveArena [Yang et al., 2025] simulator. Specifically, models are trained on Bench2Drive (source) and evaluated zero-shot on DriveArena (target). We report Route Completion (RC↑) and PDMS↑, which measure task success and planning quality in closed-loop environments. Table 7 shows that RoCA consistently improves performance over the corresponding base planners. For example, RoCA-Orion improves route completion from 17.9 to 33.7 and PDMS from 71.6 to 91.2, demonstrating substantial gains in interactive driving performance under domain

shift. Importantly, RoCA is plug-and-play for real-world simulators such as DriveArena. The Gaussian Process module is used only during training and adaptation, and the deployed planner remains unchanged, incurring no additional inference latency.

### 4.6. Cross-City Evaluation

Table 3 summarizes the results of transferring models across cities for evaluating cross-domain performance. When performing zero-shot inference in the target city without adaptation, RoCA performs more robustly as compared to the baseline. For instance, using VAD-Tiny as the baseline and running Boston-trained models in Singapore, the collision rate of RoCA is less than half of VAD-Tiny (0.211% vs. 0.430%). When adapting with ground truth, RoCA significantly outperforms direct finetuning across all metrics. For example, on Boston → Singapore transfer using SparseDrive, RoCA reduces L2 error from 0.55 m to 0.49 m and collision rate from 0.12% to 0.09%. This setting provides the fairest comparison, as both approaches utilize ground-truth supervision. Even when unsupervised, RoCA still outperforms direct finetuning with ground truth in most cases, highlighting its strong domain adaptation capability and the benefit of our probabilistic modeling.

### 4.7. Active Learning

In the target domain, active learning reduces annotation and adaptation costs if the the most informative samples are identified and used for training. To achieve this, we propose using the GP-based predictive variance as a uncertainty criterion, selecting samples with the highest estimated variance. We compare our variance-based selection with the random sampling baseline, evaluated at 5%, 10%, and 15% sampling rates of the full target training data. Table 4 reports cross-city transfer results between Singapore and Boston after finetuning using ground-truth supervision, with SparseDrive-S as the base model. Across all sampling rates, uncertainty-based selection with RoCA consistently improves planning performance over random sampling. Furthermore, RoCA consistently outperforms the baseline under both sampling strategies, underscoring its better adaptation capability.

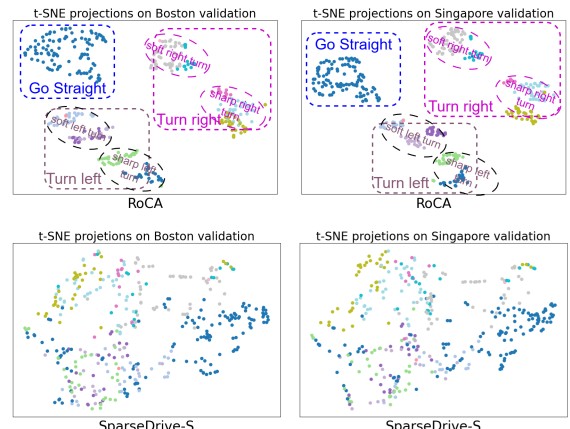

*Figure 3.* tSNE projection of ego/agent tokens with (top) and without (bottom) RoCA. By using our proposed approach, the model has better separability of different trajectory modes (indicated by different colors). In contrast, the baseline SparseDrive shows poor separability, indicating a sensitivity to any perturbations. The analysis is performed on the full nuScenes val set, the Boston and Singapore subsets(left, middle, right). Models are trained on a source city and evaluated using t-SNE projections on a target city. For example, in the left column, the model is trained on Singapore and plotted on Boston validation.

*Table 8.* Active learning comparison with different uncertainty estimation methods.

| Adapt Method | Sampling | S→B (10%) | S→B (15%) | B→S (10%) | B→S (15%) |
|---|---|---|---|---|---|
| Direct finetune | MC Dropout | 0.729 | 0.694 | 0.814 | 0.806 |
| Direct finetune | Deep Ensemble | 0.708 | 0.683 | 0.802 | 0.781 |
| RoCA | Deep Ensemble | 0.577 | 0.539 | 0.641 | 0.608 |
| RoCA | RoCA | **0.554** | **0.513** | **0.604** | **0.561** |

**Comparison with uncertainty estimation approaches.** To further validate the effectiveness of the proposed GP-based uncertainty estimation, we extend our analysis by comparing against stronger uncertainty-aware sampling baselines beyond random selection. Specifically, we consider MC Dropout and Deep Ensembles [Lakshminarayanan et al., 2017] as representative approaches for uncertainty estimation. For a fair comparison, we evaluate these methods under the same cross-domain active learning setup described in Section 4.5, considering both transfer directions (Singapore → Boston and Boston → Singapore) and different labeling budgets (10% and 15% of the target data). We report the average L2 trajectory error as the primary metric. As shown in Table 8, RoCA with GP-based uncertainty achieves the lowest L2 error across all settings, demonstrating superior uncertainty calibration for active learning under domain shift.

### 4.8. Learned Tokens in GP

Our GP-based formulation in RoCA yields a more structured and semantically meaningful token space for ego and agent trajectories, leading to robust generalization across domains. As illustrated in Figure 3, RoCA produces well-separated clusters that correspond to distinct trajectory modes, whereas the baseline SparseDrive-S exhibits significant overlap and entanglement, making it sensitive to

perturbations or noises to tokens to this token (*e.g.,* due to different camera characteristics, lighting, etc.). The improved separability is further evident in Figure 4, where commands like *Go Straight*, *Turn Left*, and *Turn Right* form coherent groups, with sub-clusters capturing fine-grained variations (e.g., soft vs. sharp turns). This organization arises from the GP's kernel-based similarity structure and its uncertainty-aware regularization, enabling RoCA to maintain stable behavior and infer consistent trajectories even under domain shift.

### 4.9. Adaption to Image Degradations

We further assess the generalization and adaptation performance when the image quality is compromised, *e.g.,* due to low light, motion blur, snow, and fog. Table 5 shows that, without adaptation, the baseline SparseDrive-S has significantly worse performance under such adverse conditions, whereas the model trained with our RoCA regularization performs more robustly. When adaptation is performed, our RoCA unsupervised adaptation achieves significant improvement, and is on par with or better than the baseline adaptation which trains the model using ground-truth data with the adverse-conditioned images. When we further utilize the ground truth in adaptation, RoCA achieves significantly better planning performance. These results confirm that RoCA strengthens generalization and adaptation under diverse degradations, improving the reliability of E2E planning.

### 4.10. Qualitative Results

Figure 2 shows qualitative planning results on turning scenarios. Our RoCA approach generates trajectories that closely align with the ground-truth trajectories. On the other hand, the baseline SparseDrive model produces trajectories that will lead the vehicle into non-drivable areas.

## 5. Conclusions

We introduced RoCA, a robust cross-domain E2E autonomous driving framework built on a Gaussian Process formulation that jointly models ego and agent trajectories. This probabilistic structure enables uncertainty-aware planning, improves source-domain training through GP-based regularization, and significantly enhances generalization to unseen environments. A key strength of RoCA is its flexible adaptation capability, supporting standard finetuning, uncertainty-guided active learning, and online adaptation for practical deployment.

## Impact Statement

RoCA improves the robustness of E2E planning models, contributing toward safer autonomous driving. Its data-efficient adaptation mechanisms reduce annotation and training requirements, potentially lowering both development time and associated energy costs.

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

# A. nuScenes Evaluation

## A.1. Evaluation protocol

We evaluate the predicted trajectory of the ego vehicle for 3 seconds into the future, with 2 waypoints per second. We use two metrics, the L2 error (in meters) between the predicted and ground-truth waypoints, and the collision rate (in %) between the ego-vehicle and the surrounding vehicles. Since the evaluation protocols in earlier works [Hu et al., 2023; Jiang et al., 2023; Zhai et al., 2023] were not consistent, *e.g.,* using different BEV grid sizes, not handling invalid/noisy frames, we adopt the standardized evaluation proposed by [Weng et al., 2024], which provides a fair comparison across methods.

## A.2. Standardized Evaluation on nuScenes Validation

Table 9 summarizes the results on nuScenes validation set, using the standardized evaluation protocol by [Weng et al., 2024]. We see that our proposed method RoCA significantly improves the baseline model and achieves competitive performance among the latest state of the art. For instance, when using VAD-Tiny as the base model, by imposing RoCA's training regularization, we reduce the average L2 error by 16% and collision rate by 38%. Note that this improvement solely results from our improved training and does not require any additional computation at inference time.

When using the trained Gaussian process module in RoCA to predict trajectory, we achieve further improved planning performance, for both base models of VAD-Tiny and SparseDrive-S, reducing the collision rate by 54% and 36%, respectively. This mode of operation, however, requires running the GP module along with the base E2E model and thus, incurs extra inference cost.

While the motivation of RoCA is mainly to enhance cross-domain performance, it is able to boost the planner's performance even when training and testing are performed in the same domain.

*Table 9.* Comparison of L2 trajectory error and collision rate comparison on nuScenes validation set [Caesar et al., 2020] using the standardized evaluation [Weng et al., 2024]. *Uses the trajectory inferred by the RoCA Gaussian process module.

| Model | Using Ego Status | Full Val | | Targeted Val | |
|---|---|---|---|---|---|
| | | Avg. L2 (m)↓ | Avg. Col. (%)↓ | Avg. L2 (m)↓ | Avg. Col. (%)↓ |
| UniAD [Hu et al., 2023] | ✗ | 0.95 | 0.45 | 1.59 | 0.47 |
| PARA-Drive [Weng et al., 2024] | ✗ | 0.66 | 0.26 | 1.08 | 0.24 |
| TOKEN [Tian et al., 2025] | ✗ | 0.68 | 0.15 | – | – |
| VAD-Tiny [Jiang et al., 2023] | ✗ | 0.91 | 0.39 | 1.27 | 0.39 |
| w/ RoCA training regularization | ✗ | 0.76 | 0.24 | 0.99 | 0.28 |
| w/ RoCA trajectory prediction* | ✗ | 0.64 | 0.18 | 0.91 | 0.25 |
| AD-MLP [Zhai et al., 2023] | ✓ | 0.66 | 0.28 | 1.13 | 1.40 |
| TOKEN [Tian et al., 2025] | ✓ | 0.64 | 0.13 | – | – |
| PARA-Drive+ [Weng et al., 2024] | ✓ | 0.59 | 0.19 | 0.70 | 0.24 |
| SparseDrive-S [Sun et al., 2025] | ✓ | 0.65 | 0.14 | 0.85 | 0.31 |
| w/ RoCA training regularization | ✓ | 0.63 | 0.13 | 0.77 | 0.28 |
| w/ RoCA trajectory prediction* | ✓ | 0.55 | 0.09 | 0.65 | 0.25 |

## A.3. RoCA Training details for this experiment

We trained RoCA using 2 NVIDIA A100 GPUs. As described in Section 3.3 of main paper, in the source domain, we first train the base E2E model following standard supervised procedure, for 48 epochs, requiring approximately 24 hours. Then, we train the Gaussian process in RoCA for 6 epochs, taking approximately 3 hours, to learn the basis tokens, based on the loss in Eqs. 3 and 5 of main paper. Finally, we finetune the base model using the standard supervised trajectory loss and RoCA regularization of Eq. 6 of the main paper for 20 epochs, taking around 10 hours. When adapting in the target domain, using the GP module as the teacher, we finetune the base model for 10 epochs, which takes approximately 5 hours.

Algorithm 1 provides the pseudo code of our source-domain training (showing 1 epoch for each step). Algorithm 2 provides the pseudo code of our target-domain unsupervised adaptation (showing 1 epoch for each step)

---

**Algorithm 1** RoCA source-domain training

---

1: Training samples in source domain: $\mathcal{D}_s = \{S_1^s, \ldots, S_N^s\}$
2: In base E2E model, tokenizer $st(.)$ with parameters $\theta_{st}$ and planner $h(.)$ with parameters $\theta_h$
3: In RoCA module, Gaussian process $g(.)$ with parameters $\theta_g$, basis token codebook $\mathcal{B} = \{\mathbf{B}_k = \{b_{j,k}\}_{j=1}^C\}_{k=1}^{N_{code}}$, and
   trajectory codebook $\mathcal{W} = \{\mathbf{W}_k = \{w_{j,k}\}_{j=1}^C\}_{k=1}^{N_{code}}$

   *# Step 1: Pretrain base E2E model*
4: **for** $S_i^s \in \mathcal{D}_s$ **do**
5:    e, a $= st(S_i^s)$    *# extract ego and agent tokens*
6:    $p_{pred}$, $c_{pred}$, $p_{pred,a}$, $c_{pred,a} = h(e, a; \theta_h)$    *# predicting ego and agent trajectories*
7:    Compute standard trajectory losses: $\mathcal{L}_{plan}(p_{pred}, p_{GT})$ and $\mathcal{L}_{mot}(p_{pred,a}, p_{GT,a})$, as in [Sun et al., 2025; Jiang et al., 2023]
8:    Update $\theta_{st}$ and $\theta_h$ using $\mathcal{L}_{plan}(p_{pred}, p_{GT})$ and $\mathcal{L}_{mot}$
9: **end for**

   *# Step 2: Learn basis tokens and RoCA parameters*
10: **for** $S_i^s \in \mathcal{D}_s$ **do**
11:    Keep $\theta_{st}, \theta_h$ frozen
12:    Calculate group labels $c_e$ and $c_a$ for ego and agent tokens, respectively, based on Section 3.2.2
13:    Calculate reconstructed ego and agent tokens as well as the predictive variances: $\hat{e}, \sigma_e^2, \hat{a}, \sigma_a^2$, according to Eq 3
14:    Compute loss $\mathcal{L}_{rec}$ based on Eq. 4
15:    Calculate predicted trajectories and variances for ego and agent: $\hat{p}_w, \sigma_w^2, \hat{p}_{w,a}, \sigma_{w,a}^2$, based on Eq. 5
16:    Compute loss $\mathcal{L}_{sup}$ based on Eq. 6
17:    Update $\mathcal{B}$ and $\theta_g$ using losses $\mathcal{L}_{rec}$ and $\mathcal{L}_{sup}$
18: **end for**

   *# Step 3: Finetune base E2E model with RoCA regularization*
19: **for** $S_i^s \in \mathcal{D}_s$ **do**
20:    Keep $\mathcal{B}$ and $\theta_g$ frozen
21:    Compute standard trajectory losses: $\mathcal{L}_{plan}(p_{pred}, p_{GT})$ and $\mathcal{L}_{mot}(p_{pred,a}, p_{GT,a})$, as in [Sun et al., 2025; Jiang et al., 2023]
22:    Compute loss $\mathcal{L}_{gp}$ based on Eq. 7
23:    Update $\theta_{st}$ and $\theta_h$ using both the standard trajectory losses and $\mathcal{L}_{gp}$
24: **end for**

---

---

**Algorithm 2** RoCA target-domain unsupervised adaptation

---

1: Training samples in source domain: $\mathcal{D}_{\mathbb{T}} = \{S_1^{\mathbb{T}}, \ldots, S_N^{\mathbb{T}}\}$
2: In base E2E model, tokenizer $st(.)$ with parameters $\theta_{st}$ and planner $h(.)$ with parameters $\theta_h$
3: In RoCA module, Gaussian process $g(.)$ with parameters $\theta_g$, basis token codebook $\mathcal{B} = \{\mathbf{B}_k = \{b_{j,k}\}_{j=1}^C\}_{k=1}^{N_{code}}$, and
   trajectory codebook $\mathcal{W} = \{\mathbf{W}_k = \{w_{j,k}\}_{j=1}^C\}_{k=1}^{N_{code}}$

   *# Step: Finetune base E2E model with RoCA unsupervised adaptation*
4: **for** $S_i^{\mathbb{T}} \in \mathcal{D}_{\mathbb{T}}$ **do**
5:    Keep $\mathcal{B}$ and $\theta_g$ frozen
6:    $p_{pred}$, $c_{pred}$, $p_{pred,a}$, $c_{pred,a} = h(e, a; \theta_h)$    *# predicting ego and agent trajectories*
7:    Calculate predicted trajectories and variances for ego and agent: $\hat{p}_w, \sigma_w^2, \hat{p}_{w,a}, \sigma_{w,a}^2$, based on Eq. 5
8:    Compute loss $\mathcal{L}_{gp}$ based on Eq. 7
9:    Update $\theta_{st}$ and $\theta_h$ using both the standard trajectory losses and $\mathcal{L}_{gp}$
10: **end for**

---

## A.4. Computation Analysis

We compare the computational costs without and with using our proposed RoCA module for trajectory prediction. As shown in Table 10, using RoCA to perform trajectory prediction introduces slightly increased latency and parameters. However, this also considerably improves planning performance, as we have seen in Table 9.

*Table 10.* Latency, inferences per second (IPS), and parameters for VAD-Tiny and SparseDrive-S without and with RoCA Gaussian process-based trajectory prediction. These measurements are conducted on an NVIDIA GeForce RTX 3090 GPU.

| Metrics | RoCA (VAD-Tiny) | | RoCA (SparseDrive-S) | |
|---|---|---|---|---|
| | Base model | Base model + RoCA | Base model | Base model + RoCA |
| Latency (ms) | 59.5 | 75.7 | 133 | 167 |
| IPS | 16.8 | 13.2 | 7.5 | 6 |
| Parameters (M) | 15.4 | 16.8 | 88.5 | 89.9 |

## A.5. Ablation Study

We conduct an ablation study to analyze the contribution of each loss component used during source-domain training. All experiments use SparseDrive-S as the base E2E model. The full configuration (ID 5 in Table 11) corresponds to the "SparseDrive-S w/ RoCA training regularization" entry reported earlier in Table 9 of the main paper.

Table 11 summarizes the results. Starting from a model trained without any of the additional loss terms (ID 1), we observe that enabling the reconstruction loss $\mathcal{L}_{rec}$ alone (ID 2) already leads to a reduction in both trajectory error and collision rate. Adding the core supervision losses within $\mathcal{L}_{sup}$ (ID 3) further improves accuracy, confirming their importance in guiding the planner toward consistent future trajectory predictions.

Incorporating the classification term $\mathcal{L}_{class}$ (ID 4) yields additional gains by encouraging more stable behavioral mode selection. Finally, including the triplet regularization term $\mathcal{L}_{tripplet}$ (ID 5) provides the best overall performance, producing lower L2 error and fewer collisions than all preceding variants.

### A.5.1. ABLATION STUDY ON KERNEL FUNCTIONS

We conducted an experiment to compare using different kernel functions, including radial basis function (RBF), linear kernel (Lin), and rotational quadratic kernel (RQ). As shown in Table 12 below, the different choices of kernels (i.e., pairwise similarity functions) do not significantly affect the performance.

### A.5.2. ABLATION STUDY ON $N_{ego}$

To understand the effect of diversity, we compare using different numbers of groups, i.e., 6 and 16, for each driving command, as shown in Table 13 here. It can be seen that using fewer groups (i.e., less diversity) can result in worse planning performance as trajectory diversity is lacking.

## B. Comparison with different planners

Existing deterministic approaches based on MLP planner heads or anchor-based planner heads (e.g., k-means or deterministic residual prediction) like diffusion planner heads, treat trajectory anchors as fixed prototypes and rely on hard classification followed by residual regression. This approach does not account for uncertainty and cannot adaptively weigh predictions when encountering out-of-distribution scenarios. In contrast, RoCA models the joint distribution over token embeddings and trajectories using a GP formulation provides two key advantages: (i) **Kernel-based similarity** ensures that predictions are influenced by all relevant basis tokens rather than a single nearest anchor, and (ii) **Uncertainty-aware inference** via $\sigma^2$ enables RoCA to regularize training and adapt to ambiguous or unseen scenarios. To further isolate the benefit of GP, in Table 14 we have added an comparing RoCA with a deterministic anchor-based variant example diffusion planner. The GP variant consistently outperforms the deterministic version, validating the advantage of our approach.

Table 11. Effect of different loss terms on source-domain training.

| ID | $\mathcal{L}_{rec}$ | $\mathcal{L}_{sup}$ | | | Avg. L2 (m) ↓ | Avg. Col. (%) ↓ |
|----|---------------------|---------------------|---------------------|-----------------------|---------------|-----------------|
| | | $\mathcal{L}_{plan} + \mathcal{L}_{mot}$ | $\mathcal{L}_{class}$ | $\mathcal{L}_{tripplet}$ | | |
| 1 | ✗ | ✗ | ✗ | ✗ | 0.65 | 0.140 |
| 2 | ✓ | ✗ | ✗ | ✗ | 0.65 | 0.136 |
| 3 | ✓ | ✓ | ✗ | ✗ | 0.64 | 0.130 |
| 4 | ✓ | ✓ | ✓ | ✗ | 0.63 | 0.130 |
| 5 | ✓ | ✓ | ✓ | ✓ | 0.63 | 0.127 |

Table 12. Using different kernel functions for $\kappa$ in the Gaussian process module, with SparseDrive-S as the base model and evaluating on nuScenes.

| Metric | Lin | RBF | RQ |
|--------|------|------|------|
| Avg L2 | 0.57 | 0.55 | 0.52 |
| Avg Col | 0.09 | 0.09 | 0.09 |

Table 13. Ablation study on different $N_{ego}$ groups used to construct the trajectory codebook. We use SparseDrive-S as the base model and evaluate on nuScenes.

| Metric | $N_{ego} = 3*6$ | $N_{ego} = 3*16$ |
|--------|------------------|-------------------|
| Avg L2 | 0.58 | 0.55 |
| Avg Col | 0.10 | 0.09 |

Table 14. Comparison with diverse planners. DS and SR denote Driving Score and Success Rate respectively.

| Generative Planner | Closed-loop | | Open-loop | Ability Avg. |
|--------------------|-------|--------|-----------------|--------------|
| | DS↑ | SR(%)↑ | Avg. L2 (m)↓ | |
| ORION w/ MLP | 70.73 | 41.52 | 0.75 | 48.44 |
| ORION w/ Diffusion | 71.97 | 46.54 | 0.73 | 46.68 |
| ORION w/ VAE | 77.74 | 54.62 | 0.68 | 54.72 |
| RoCA (ORION) w/ GP | 80.38 | 58.22 | 0.57 | 61.11 |

## C. Gaussian Processes

A Gaussian process (GP) $f(v)$ is as an infinite collection of random variables, where any finite subset follows a joint Gaussian distribution. A GP is fully characterized by its mean function and covariance function, given by

$$m(v) = \mathbb{E}[f(v)], \tag{8}$$

$$K(v, v') = \mathbb{E}\big[(f(v) - m(v))(f(v') - m(v'))\big], \tag{9}$$

where $v, v' \in \mathcal{V}$ represent possible input points. The covariance matrix is derived from a kernel function $K$, which encodes prior assumptions about the smoothness of the underlying function. A GP can then be expressed as

$$f(v) \sim \mathcal{GP}(m(v), K(v, v') + \sigma_\epsilon^2 I), \tag{10}$$

where $I$ is the identity matrix and $\sigma_\epsilon^2$ denotes the variance of additive noise. For a set of inputs $V = \{v_1, \ldots, v_n\}$, the corresponding function values follow

$$f(V) = [f(v_1), \ldots, f(v_n)]^T \sim \mathcal{N}(\mu, K(V, V') + \sigma_\epsilon^2 I), \tag{11}$$

with mean vector $\mu$ and covariance matrix defined by the GP. To predict at unlabeled points, one can compute the Gaussian posterior distribution in closed form by conditioning on observed data. For a detailed review of Gaussian processes, refer to [Rasmussen, 2003].

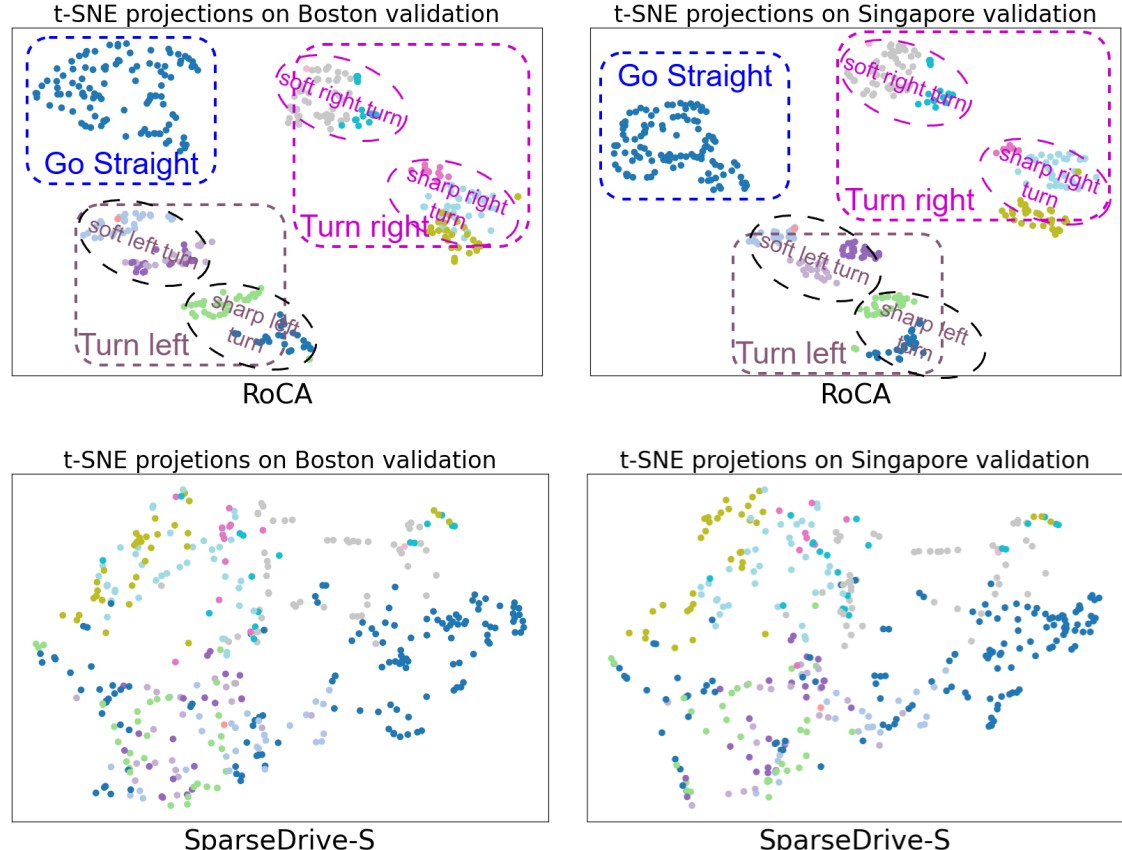

*Figure 4.* t-SNE projections of ego/agent tokens with (top) and without (bottom) RoCA. Baseline SparseDrive-S exhibits mixed and overlapping clusters, indicating weak separation and stronger sensitivity to domain shift. In contrast, RoCA leverages its Gaussian-Process–based formulation to model kernel similarities and uncertainty-aware inference, resulting in clearer clustering structure and improved robustness. Models are trained on a source city and evaluated using t-SNE projections on a target city. For example, in the left column, the model is trained on Singapore and plotted on Boston validation.

### C.1. Why Gaussian Processes Improve Robustness.

Unlike deterministic anchor-based methods, GPs model the joint distribution of token embeddings and trajectories. This formulation provides two key advantages: (i) **Kernel-based similarity** ensures that embeddings close in latent space influence predictions more strongly, and (ii) **Uncertainty-aware inference** via $\sigma^2$ enables RoCA to regularize training and adapt to ambiguous or unseen scenarios. These properties lead to tighter clusters and better generalization, as evidenced by the improved separability in Figure 4.

Figure 4 illustrates the t-SNE projections of token embeddings for different driving commands on Boston and Singapore validation sets. Compared to the baseline SparseDrive, our proposed RoCA method produces well-clustered and clearly separated groups corresponding to commands such as *Go Straight*, *Turn Left*, and *Turn Right*. Within the *Turn Left* group, sub-clusters representing sharp left turns are farther from the *Go Straight* cluster, while very soft left turns are closer—reflecting semantic similarity in driving behavior. This improved separation stems from RoCA's probabilistic formulation using Gaussian Processes (GPs).

## D. Comparison with VLP [Pan et al., 2024]

VLP is one of few existing works that investigate cross-domain performance for end-to-end autonomous driving. Since VLP authors have not released their codes/models, we cannot evaluate their method using the standardized nuScenes evaluation protocol. In order to compare with their reported numbers in the paper, here we use the VAD [Jiang et al., 2023] evaluation protocol.

Table 15 summarizes the comparison of cross-domain generalization performance on nuScenes validation set. The models

are trained on the subset of training data collected in one city (*e.g.*, Boston) and then, zero-shot evaluated on the validation data belonging to the other city (*e.g.*, Singapore). We see that our proposed RoCA provides the best domain generalized planning.

Table 15. Cross-domain evaluation on Boston and Singapore subsets of nuScenes validation.

| Model | Boston | | Singapore | |
|---|---|---|---|---|
| | Avg. L2 (m) ↓ | Avg. Col. (%) | Avg. L2 (m) ↓ | Avg. Col. (%) |
| VAD-Tiny [Jiang et al., 2023] | 0.86 | 0.27 | 0.78 | 0.39 |
| SparseDrive-S [Sun et al., 2025] | 0.84 | 0.23 | 0.70 | 0.15 |
| VLP (VAD) [Pan et al., 2024] | 0.73 | 0.22 | 0.63 | 0.20 |
| VLP (UniAD) [Pan et al., 2024] | 1.14 | 0.26 | 0.87 | 0.34 |
| RoCA (VAD-Tiny) | 0.69 | 0.14 | 0.56 | 0.21 |
| RoCA (SparseDrive-S) | 0.52 | 0.09 | 0.50 | 0.12 |

## E. Cross-City Evaluation on Targeted Validation Split

As an extension of Table 3 in the main paper, Table 16 here provides the cross-domain performance on targeted scenarios (where the ego vehicle has to make a turn) on nuScenes.

We see that in these more challenging scenarios, our proposed RoCA consistently outperforms the baseline model in terms of domain generalization (no adaptation), and the direct finetuning in terms of domain adaptation (model weights are updated). It is noteworthy that even without using the ground-truth trajectory annotations, models adapted using RoCA perform better than those directly finetuned with ground truth in the target city. When using ground-truth labels, RoCA further improves the performance of the adapted models.

Table 16. Cross-city planning performance on targeted scenarios.

| Method | Adapt | Boston → Singapore | | Singapore → Boston | |
|---|---|---|---|---|---|
| | | Avg. L2 (m) ↓ | Avg. Col.(%) ↓ | Avg. L2 (m) ↓ | Avg. Col.(%) ↓ |
| VAD-Tiny | ✗ | 1.629 | 0.482 | 1.157 | 0.224 |
| w/ finetuning using GT | ✓ | 1.402 | 0.270 | 1.105 | 0.190 |
| w/ RoCA training regularization | ✗ | 1.241 | 0.256 | 1.006 | 0.198 |
| w/ RoCA unsupervised adaptation | ✓ | 1.035 | 0.231 | 0.951 | 0.172 |
| w/ RoCA adaptation using GT | ✓ | 0.897 | 0.200 | 0.890 | 0.169 |
| SparseDrive-S | ✗ | 1.061 | 0.281 | 1.014 | 0.241 |
| w/ finetuning using GT | ✓ | 0.785 | 0.199 | 0.671 | 0.122 |
| w/ RoCA training regularization | ✗ | 0.942 | 0.192 | 0.886 | 0.156 |
| w/ RoCA unsupervised adaptation | ✓ | 0.800 | 0.104 | 0.688 | 0.115 |
| w/ RoCA adaptation using GT | ✓ | 0.701 | 0.094 | 0.511 | 0.104 |

## F. Long-Tail Evaluation on nuScenes Validation

Table 17 is an extension of Table 9 in the main paper, where we evaluate the long-tail scenarios, such as resume from step, overtake, and 3-point turn, in the nuScenes validation set, using the standardized evaluation protocol. We use the average L2 trajectory error as the metric and do not use the collision rate here, as it is less statistically stable due to the small number of long-tail samples.

We see that RoCA enables better performance in these long-tail scenarios, confirming its effectiveness in making the model more generalizable. In particular, while a base model like VAD-Tiny under-performs other existing methods, by leveraging RoCA, we are able to significantly boost its performance and make it work better in these challenging long-tail cases.

*Table 17.* Comparison of L2 trajectory error and collision rate comparison on long-tail scenarios of nuScenes validation set [Caesar et al., 2020] using the standardized evaluation [Weng et al., 2024]. *Uses the trajectory inferred by the RoCA Gaussian process module.

| Model | Using Ego Status | Resume from Stop | Overtake | 3-Point Turn |
|---|---|---|---|---|
| PARA-Drive [Weng et al., 2024] | ✗ | 1.08 | 1.03 | 1.55 |
| TOKEN [Tian et al., 2025] | ✗ | 0.80 | 0.90 | 1.43 |
| VAD-Tiny [Jiang et al., 2023] | ✗ | 1.75 | 1.32 | 1.83 |
| w/ RoCA training regularization | ✗ | 1.13 | 1.21 | 1.72 |
| w/ RoCA trajectory prediction* | ✗ | 0.95 | 1.00 | 1.41 |
| TOKEN [Tian et al., 2025] | ✓ | 0.65 | 0.74 | 0.73 |
| Senna [Jiang et al., 2024] | ✓ | 1.44 | 0.94 | 1.59 |
| DiMA [Hegde et al., 2025] | ✓ | 1.11 | 0.82 | 1.27 |
| SparseDrive-S [Sun et al., 2025] | ✓ | 0.67 | 0.69 | 0.79 |
| w/ RoCA training regularization | ✓ | 0.41 | 0.63 | 0.71 |
| w/ RoCA trajectory prediction* | ✓ | 0.34 | 0.53 | 0.61 |

## G. Motion prediction evaluation on nuScenes validation

We evaluate the agent trajectories using minADE (minimum Average Displacement Error) and minFDE (minimum Final Displacement Error) metrics in Table 18 below with and without using RoCA. Here we use SparseDrive-S as the base model. It can be seen that RoCA clearly improves agent trajectory prediction when used as a training regularization.

*Table 18.* Agent trajectory evaluation on nuScenes dataset. Both minADE and minFDE are lower the better. *Uses the trajectory inferred by the RoCA Gaussian process module.

| Method | minADE ↓ | minFDE ↓ |
|---|---|---|
| SparseDrive-S | 0.62 | 0.99 |
| SparseDrive-S w/ RoCA training regularization | 0.54 | 0.83 |
| SparseDrive-S w/ RoCA trajectory prediction* | 0.51 | 0.77 |

## H. Training plots

### H.1. Variance plot across epochs

In practice, the RoCA framework assigns higher $\sigma^2$ values to problematic or out-of-distribution token embeddings and gradually reduces these values as training progresses across epochs. Figure 5 illustrates the variance across epochs during RoCA training regularization on the source dataset using Eq. 6 from the main paper for 20 epochs on the nuScenes dataset.

Table 19 clearly shows that RoCA is slightly less confident in predicting way-point trajectories and has higher variance values compared to nuScenes validation. Similarly, variance values for long-tail scenes like "Overtake" and "Resume from stop" are higher than nuScenes validation, indicating that the confidence of RoCA is lower in these cases.

*Table 19.* Variance values for nuScenes validation and long-tail scenarios including Resume from stop, Overtake, and 3-point Turn.

| Variance | Val | Long-tail | | |
|---|---|---|---|---|
| | | Resume from stop | Overtake | 3-point Turn |
| | 0.87 | 1.08 | 1.21 | 1.56 |

### H.2. Token embeddings plot across epochs

Figure 6 illustrates the t-SNE projections of 500 "Turn left" and 500 "Turn right" driving command of token embeddings across training epochs (1, 3, 6, and 18). The plots illustrate the progressive clustering of embeddings as training advances. Early epochs (1 and 3) show dispersed clusters, while later epochs (6 and 18) exhibit well-separated groups corresponding to driving maneuvers. By epoch 18, clusters representing critical actions such as *left turn* and *right turn* become clearly distinguishable, indicating improved semantic organization in the learned representation.

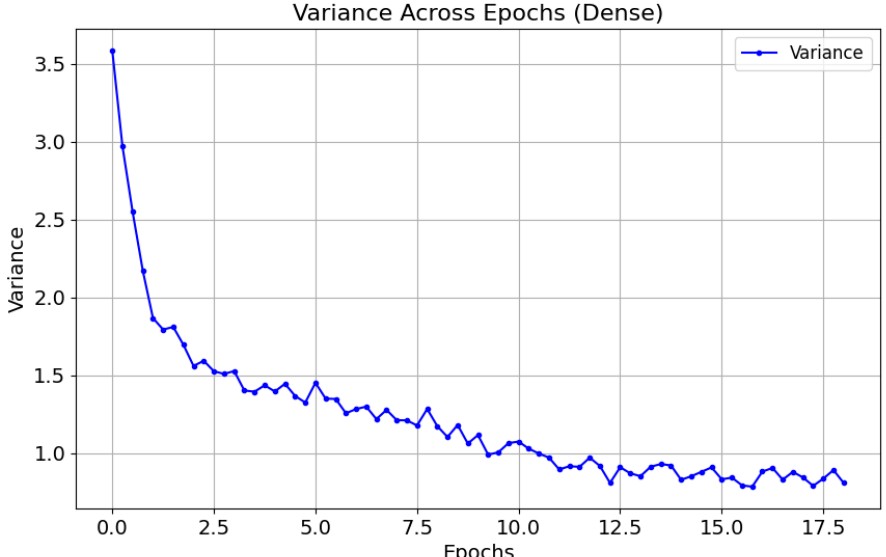

*Figure 5.* Variance across epochs during RoCA training regularization on source dataset using Eq. 6 of the main paper for 20 epochs, on nuScenes dataset.

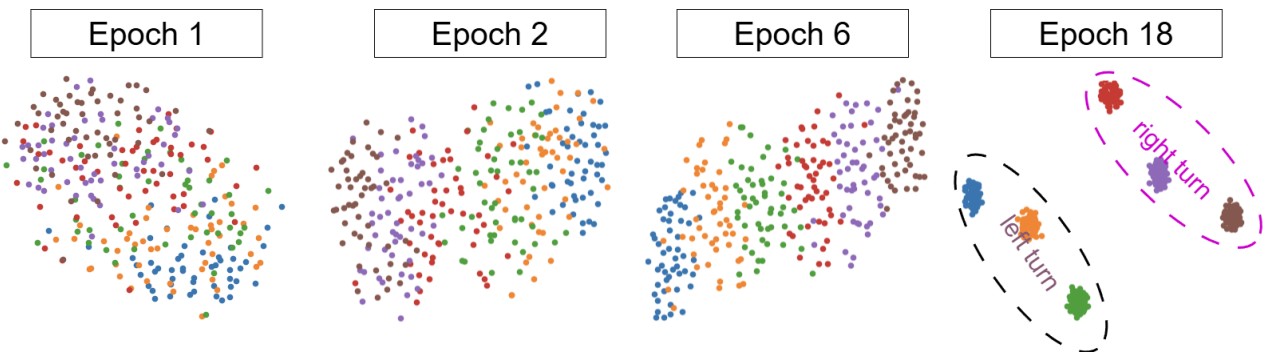

*Figure 6.* t-SNE visualization of token embeddings across training epochs (1, 3, 6, and 18). The plots illustrate the progressive clustering of embeddings as training advances. Early epochs (1 and 3) show dispersed clusters, while later epochs (6 and 18) exhibit well-separated groups corresponding to driving maneuvers. By epoch 18, clusters representing critical actions such as *left turn* and *right turn* become clearly distinguishable, indicating improved semantic organization in the learned representation.

## I. Additional Qualitative Results

We show additional qualitative results on challenging scenes in the nuScenes validation set (best viewed in color).[3]

Fig. 7 shows a night-time scenario where it is very dark. RoCA generates the correct trajectory while the baseline leads the car outside the drivable area.

Fig. 8 shows a sample at an intersection with several other cars and multiple pedestrians. RoCA provides the right action. The baseline generates a trajectory that goes out of the road. The predicted trajectories of other agents are also shown in the figure.

---

[3]Images from nuScenes, licensed under .

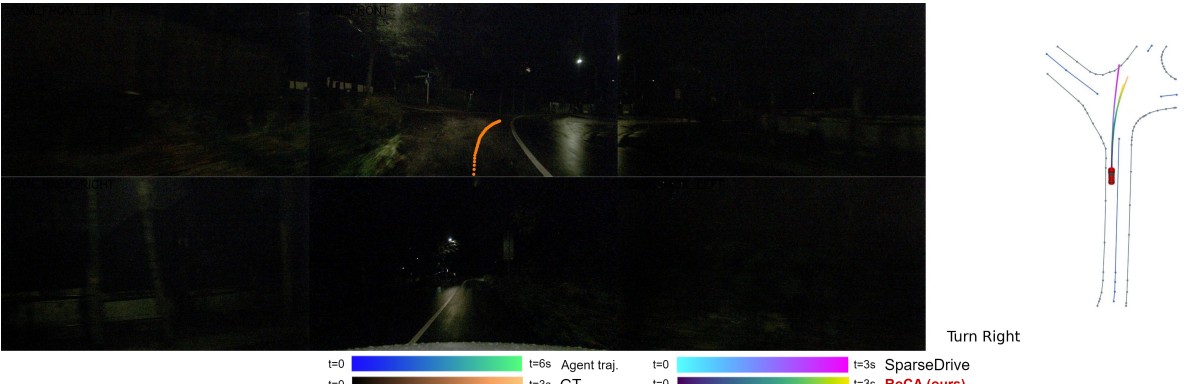

*Figure 7.* Qualitative result on a night-time scenario in nuScenes validation set.

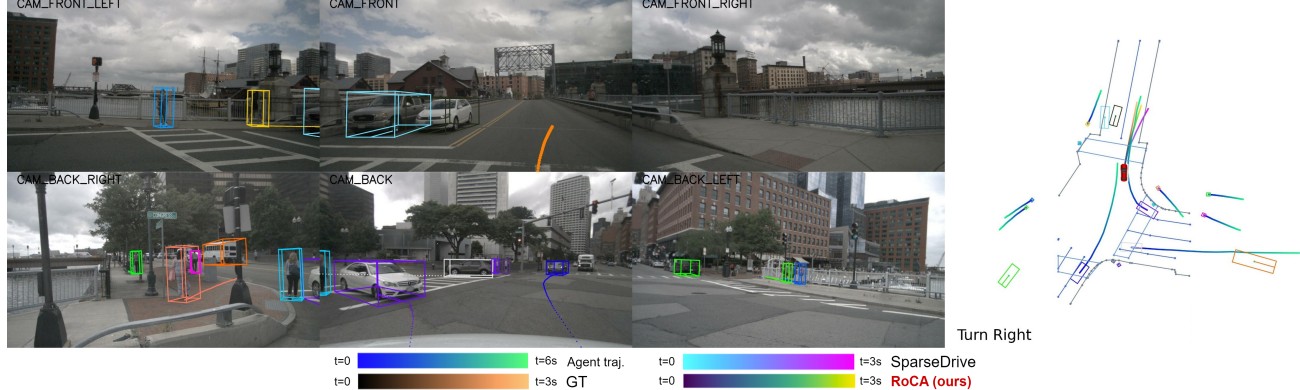

*Figure 8.* Qualitative result on an intersection scenario in nuScenes validation set.

