# OpenReview forum: "RoCA: Robust Cross-Domain End-to-End Autonomous Driving"
_ICML.cc/2026/Conference — ICML 2026 regular_

### Official Review · Reviewer_bApd · 2026-03-05

**Soundness:** 3
**Presentation:** 2
**Significance:** 2
**Originality:** 2
**Overall Recommendation:** 4
**Confidence:** 4

**Summary:**

The paper introduces ROCA, a framework designed to enhance the robustness and cross-domain adaptation of end-to-end (E2E) autonomous driving systems. To address the performance degradation when deploying models in unseen environments (different cities or weather conditions), the authors propose a Gaussian Process (GP)-based approach. ROCA maps scene tokens to a structured latent space using a "basis token" codebook and formulates trajectory prediction as a probabilistic inference problem. This allows the model to leverage uncertainty estimation for both robust prediction and efficient active learning during domain adaptation. The framework is designed to be "plug-and-play" with existing E2E models like VAD or SparseDrive. Experiments are conducted on nuScenes (open-loop) and CARLA/Bench2Drive (closed-loop) to demonstrate its efficacy in cross-domain scenarios.

**Compliance With Llm Reviewing Policy:**

Affirmed.

**Ethical Review Flag:**

Flag this paper for an ethics review.

**Final Justification:**

Most of my concerns have been satisfactorily addressed. I will raise my score.

**Key Questions For Authors:**

How does RoCA compare against other standard techniques for domain adaptation or robust representation, such as Contrastive Learning (e.g., MoCo-based pre-training) or Bayesian Neural Networks (Monte Carlo Dropout) applied to the same E2E baselines?

The codebook is learned from the source domain (nuScenes). In a scenario where the target domain has fundamentally different road geometries or driving rules (e.g., switching from right-hand to left-hand traffic), how does the model handle "unknown" tokens that do not map well to the existing basis? Is there a fallback mechanism?

In the Bench2Drive closed-loop experiments, could you provide more qualitative or quantitative analysis on whether RoCA specifically improves reactive safety (e.g., responding to a sudden cut-in) compared to the baseline?

What is the sensitivity of the model to the number of basis tokens ($N_{code}$)? If the number of tokens is significantly reduced, does the uncertainty estimation become unreliable?

**Limitations:**

See Pros and Cons.

**Strengths And Weaknesses:**

**Strengths:**

1. **Versatility:** The "plug-and-play" nature of the ROCA module is impressive. The authors demonstrate its compatibility with multiple baseline architectures (e.g., VAD, SparseDrive), showing consistent improvements across different backbones.
2. **Effective Active Learning:** The results on unsupervised domain adaptation (UDA) are compelling, showing that using GP-based uncertainty to select informative samples can significantly reduce the amount of target-domain data needed for fine-tuning.

**Weaknesses:**

1. **Limited Technical Originality:** While the combination of GP and E2E driving is interesting, the individual components (VQ-VAE-style codebooks, GP for uncertainty, and active learning) are well-established in other computer vision and robotics domains. The paper feels more like a creative application of existing tools rather than a fundamental algorithmic breakthrough.
2. **Computational Overhead and Real-time Concerns:** A major concern in ICML-level robotics/AI papers is the trade-off between probabilistic inference and real-time performance. Although the authors mention a "regularization mode" that doesn't add inference cost, the "inference mode" (which provides the best performance) involves GP operations. The paper lacks a detailed analysis of the latency overhead on edge-computing hardware typically used in vehicles (e.g., NVIDIA Orin).
3. **Insufficient Comparison with Baselines:** The cross-domain evaluation compares ROCA-enhanced models against their vanilla counterparts. However, it lacks a robust comparison against other established domain adaptation or robust representation learning techniques in the E2E AD field (e.g., contrastive learning or standard Bayesian Neural Networks). Without these, it is difficult to isolate whether the GP specifically is the optimal solution.
4. **Clarity on Closed-loop Generalization:** While Bench2Drive results are provided, the gap between open-loop metrics (L2/Collision) and actual driving safety in complex, interactive closed-loop environments remains large. Further analysis on why ROCA helps specifically in "closed-loop" reactive behavior would strengthen the paper.

---

> ### Author Rebuttal · Authors · 2026-03-30
>
> We thank the reviewer for the detailed and thoughtful feedback, particularly for the comments on **technical originality**, **computational efficiency**, and the evaluation of **closed‑loop reactive safety across domains**. These points helped us clarify our contributions and strengthen the empirical analysis.
>
> **W1: Limited technical originality**
>
> **Response:** To the best of our knowledge, we are the first to propose a GP based planner that leverages a learned discrete trajectory codebook. This design enables the planner to serve both **(i) as a training‑time teacher (zero inference cost)** and **(ii) as a unified engine for unsupervised and active learning adaptation** in E2E autonomous driving. This joint design enables **kernel‑weighted trajectory inference and variance‑aware supervision/selection**, improving **closed‑loop cross‑domain robustness** of baseline E2E approaches.
>
> **W2: Computational overhead**
>
> **Response:** RoCA’s default deployment adds no inference overhead: the GP runs only during training/adaptation, so the planner remains unchanged (**no GP ops, no latency or parameter increase**). The optional **GP‑at‑inference path** (for improved performance) has small overhead since it operates on small matrices (e.g., **Nego × Nego, C × C**, where **Nego = 48, C = 64**). As shown in **Appendix Table 7 (p.14)**, latency deltas are modest (e.g., **59.5→75.7 ms for VAD‑Tiny; 133→167 ms for SparseDrive‑S**) and **not required for deployment**.
>
> **W3 and Q1: comparison with domain adaptation baselines**
>
> **Response:** We added comparisons with **contrastive pretraining (ORION + MoCo)** and **Bayesian NN style uncertainty (ORION + MC Dropout)** under the same **closed‑loop protocol (Bench2Drive)** with identical training budgets and planner heads. As shown below, **RoCA (ORION) outperforms both**: **DS improves from 77.8 → 80.4 (+2.6, ~+3.3%)** and **SR from 54.6 → 58.2 (+3.6, ~+6.6%)**.
>
>
> Table R7. Closed-loop comparisons on Bench2Drive dataset
> ```
> +------------------------+------+------+
> | Method                 |  DS  |  SR  |
> +------------------------+------+------+
> | ORION                  | 77.8 | 54.6 |
> | ORION + MoCo           | 78.6 | 54.9 |
> | ORION + MC Dropout     | 78.4 | 55.2 |
> | RoCA (ORION)           | 80.4 | 58.2 |
> +------------------------+------+------+
> ```
>
>
> **W4: closed-loop generalization and reactive safety**
>
> **Response:** RoCA yields larger gains on **reactive closed‑loop behaviors** than on aggregate scores. As shown in **Table 1**, RoCA improves challenging reactive skills—including **merging, overtaking, emergency braking, and give‑way**—by larger margins than overall driving score, indicating that **GP‑based uncertainty regularization benefits rapid, safety‑critical responses**. We also include **closed‑loop NAVSIM/DriveArena results (Tables R4/R5; Reviewer 5nER)**, where RoCA improves **PDMS, ego‑progress, and route‑completion**.
>
> **Q2: Handling unseen tokens under different traffic rules or geometries**
>
> **Response:** We evaluate this scenario in **Table 4** via **cross‑city adaptation between Boston (right‑hand traffic) and Singapore (left‑hand traffic)**. Poorly aligned tokens initially exhibit **higher GP uncertainty**, triggering **uncertainty‑aware adaptation**. During adaptation, **basis tokens are fine‑tuned**, and the GP updates kernel correlations to model unseen trajectory patterns. As a result, **RoCA significantly improves performance after adaptation** without a fallback module. **t‑SNE visualizations (Figure 3)** show target‑domain tokens becoming well aligned with the adapted codebook.
>
> **Q3: Impact of RoCA on reactive safety in closed-loop settings**
>
> **Response:** RoCA improves **reactive safety behaviors** in closed‑loop settings. From **Table 1**, compared to ORION, **RoCA (ORION)** improves **overtaking (71.11 → 74.91)**, **emergency braking (78.33 → 83.76)**, and **give‑way (30 → 40)**, while increasing **mean ability (54.72 → 61.11)**. Similar trends hold for **VAD and SSR backbones**, indicating stronger responses to **safety‑critical events**.
>
> **Q4: Sensitivity to the number of basis tokens**
>
> **Response:** RoCA is not highly sensitive to basis size. As shown in **Appendix Table 10 (p.15)**, reducing the codebook from **Nego = 3×16** to **3×6** causes only minor degradation: **Avg L2 error 0.55 → 0.58** and **collision rate 0.09 → 0.10**, indicating **stable uncertainty estimation**.

---

> > ### Author Rebuttal · Reviewer_bApd · 2026-04-04
> >
> > Most of my concerns have been satisfactorily addressed. I will raise my score.

---

> > > ### Author Response · Authors · 2026-04-04
> > >
> > > Thank you for taking the time to carefully review our rebuttal and the additional experiments. We sincerely appreciate your thoughtful feedback and your acknowledgment that the concerns were satisfactorily addressed. We are grateful that you found the clarifications and new results helpful and for raising your overall recommendation.

---

### Official Review · Reviewer_5nER · 2026-03-09

**Soundness:** 3
**Presentation:** 2
**Significance:** 3
**Originality:** 3
**Overall Recommendation:** 4
**Confidence:** 2

**Summary:**

This paper proposes RoCA, a robust cross-domain end-to-end autonomous driving framework. RoCA introduces a Gaussian Process module to jointly model ego- and agent-token distributions and their corresponding trajectories. By learning a set of base tokens and trajectory codebooks, it enables probabilistic prediction, uncertainty estimation, and domain adaptation, improving generalization to new scenarios without extensive retraining. Extensive experiments on Bench2Drive and nuScenes demonstrate state-of-the-art cross-domain performance.

**Compliance With Llm Reviewing Policy:**

Affirmed.

**Final Justification:**

The rebuttal has addressed my main concerns, so I'd like to maintian my original score.

**Key Questions For Authors:**

1. To W1: How does RoCA perform in real-world end-to-end autonomous driving settings, considering differences in sensor placement and other deployment factors? Have the authors considered evaluating on additional datasets such as NavSim or NavSimV2 to better demonstrate cross-domain capability?

2. To W2: Has RoCA been tested in closed-loop real-world simulations? For example, how does it integrate with real-world closed-loop simulators or frameworks like DriveArena, which combine world models or ground truth feedback?

**Limitations:**

Yes

**Strengths And Weaknesses:**

Strengths:

1.Introduces Gaussian Processes over a learned token/trajectory codebook for trajectory prediction, allowing the model to quantify uncertainty in a principled way.

2.Improves cross-domain generalization across cities, lighting, and weather, and can be applied to existing end-to-end driving models without modifying their architecture.

3.Supports active learning, achieving strong adaptation with only a small fraction of target-domain data.

4.Shows solid experimental results on Bench2Drive and nuScenes, with ablation studies confirming the contribution of each component.

Weaknesses:

1. The cross-domain evaluation is limited to simulation datasets such as nuScenes. End-to-end autonomous driving can be sensitive to differences in sensor placement and other real-world factors, which are not explored. Previous work has shown that validation solely on nuScenes may not fully demonstrate real-world cross-domain capability. Evaluation on additional datasets like NavSim or NavSimV2 would strengthen the evidence.

2.The paper lacks closed-loop evaluation in the real world. There is no testing with real-world closed-loop simulators or frameworks, such as those integrating ground truth or world models in platforms like DriveArena. Without such evaluation, it is unclear how well the approach would perform under realistic, interactive driving conditions.

---

> ### Author Rebuttal · Authors · 2026-03-30
>
> We thank the reviewer for emphasizing the importance of **closed‑loop, cross‑domain evaluation beyond nuScenes**, particularly regarding **real‑world robustness, sensor differences, and interactive driving dynamics**. This feedback motivated us to add new closed‑loop evaluations on **NAVSIM and DriveArena**, strengthening the empirical validation of RoCA under realistic deployment conditions.
>
> **W1: Cross-domain evaluation on NAVSIM**
>
> **Response:** We agree that **open‑loop nuScenes metrics alone are insufficient** to assess real‑world robustness in cross‑domain end‑to‑end driving. To address this, we add a **closed‑loop cross‑domain evaluation** by training on **Bench2Drive (source)** and evaluating on **NAVSIM v1 (target)** using **planning‑oriented metrics**—**No‑at‑fault Collisions (NC↑)**, **Ego Progress (EP↑)**, and **PDMS↑**—which better reflect closed‑loop behavior than displacement errors.
> For **RoCA‑Orion** and **RoCA‑SparseDrive**, we further perform **unsupervised fine‑tuning** as described in **Sections 3.2.3 and 3.3.2**. As shown in **Table R4**, RoCA improves **PDMS by +3.7 over SparseDrive (85.8 vs. 82.1)** and **+3.5 over Orion (87.9 vs. 84.4)**, with concurrent gains in **EP (+2.3 / +3.2)** and **NC (+0.9 / +0.5)**. These results indicate **stronger closed‑loop planning under domain shift**, including sensitivity to **sensor configuration and interaction dynamics**. Evaluating on **NAVSIM/DriveArena‑style closed‑loop surrogates** is a meaningful step beyond nuScenes, and we plan to extend this to **NAVSIM v2** in future work.
>
> Table R4. Cross‑domain closed‑loop performance on NAVSIM v1 test (target), with Bench2Drive as source.
> Metrics: NC↑ (No‑at‑fault Collisions), EP↑ (Ego Progress), PDMS↑ (Planning quality). Higher is better.
>
> ```
> +-------------------+------+-------+-------+
> | Method            |  NC  |  EP   | PDMS  |
> +-------------------+------+-------+-------+
> | SparseDrive       | 97.1 |  74.2 |  82.1 |
> | Orion             | 97.9 |  78.7 |  84.4 |
> | RoCA-SparseDrive  | 98.0 |  77.5 |  85.8 |
> | RoCA-Orion        | 98.4 |  82.6 |  87.9 |
> +-------------------+------+-------+-------+
> ```
>
> **W2: Closed-loop evaluation in real-world simulators (DriveArena)**
>
> **Response:** We agree that **open‑loop metrics are insufficient** for evaluating performance in **interactive, real‑world settings**. To address this, we add a **closed‑loop cross‑domain evaluation** by training on **Bench2Drive (source)** and evaluating **zero‑shot on DriveArena (target)** using **Route Completion (RC↑)** and **PDMS↑**. As shown in **Table R5**, RoCA consistently improves **closed‑loop planning quality** over the corresponding base planners, indicating stronger performance under **interactive feedback**.
>
> Importantly, **RoCA is plug‑and‑play for real‑world simulators such as DriveArena**. The **Gaussian Process is used only during training/adaptation**; the deployed planner is unchanged (**no GP calls, no added latency**). Thus, integration into DriveArena—or other frameworks with world models or ground‑truth feedback—uses the **same runtime stack as the baseline planner**. The **optional GP‑at‑inference mode is not required** to obtain the reported gains.
>
>
> Table R5. Cross-domain closed‑loop surrogate evaluation (Bench2Drive → DriveArena, zero‑shot).
> Metrics: RC↑ (Route Completion) and PDMS↑ (higher is better).
>
> ```
>
> +-------------------+--------+--------+
> | Method            |   RC   |  PDMS  |
> +-------------------+--------+--------+
> | SparseDrive       |  13.7  |  68.3  |
> | Orion             |  17.9  |  71.6  |
> | RoCA-SparseDrive  |  28.1  |  84.4  |
> | RoCA-Orion        |  33.7  |  91.2  |
> +-------------------+--------+--------+
> ```

---

> > ### Author Rebuttal · Reviewer_5nER · 2026-04-02
> >
> > Thank you to the authors for the response. I will maintain my original score.

---

> > > ### Author Response · Authors · 2026-04-02
> > >
> > > Thank you for carefully reviewing our rebuttal and the additional closed‑loop evaluations. We appreciate your constructive feedback and your acknowledgment that the concerns were addressed.

---

### Official Review · Reviewer_HAAb · 2026-03-12

**Soundness:** 3
**Presentation:** 3
**Significance:** 3
**Originality:** 3
**Overall Recommendation:** 5
**Confidence:** 3

**Summary:**

The paper proposes RoCA, a framework designed to improve the cross-domain generalization capabilities of end-to-end autonomous driving models. The method leverages Gaussian Processes to cluster ego and agent tokens from the base model into trajectory-related semantic classes. These clusters are intended to capture trajectory semantics and provide additional structure to the latent representation, which in turn helps the model generalize better to novel domains. The method is evaluated in several settings, including fine-tuning and active learning scenarios, where it shows improvements over baseline approaches.

**Compliance With Llm Reviewing Policy:**

Affirmed.

**Final Justification:**

My concerns have been addressed and I have raised my score.

**Key Questions For Authors:**

[Q1] How does N_code and other parameters affect training and inference time?

[Q2] Why were the tiny (VAD-Tiny) and small (SparseDrive-S) variants of the baseline end-to-end models selected? It would be useful to clarify whether this choice could introduce any selection bias and whether similar trends hold for larger models.

[Q3] In the active learning experiments (table 4), RoCA is compared only to random sampling, which is generally easy to outperform. Have the authors considered comparing against stronger active learning baselines?

**Limitations:**

Yes/W2

**Strengths And Weaknesses:**

[S1] Improving cross-domain generalization for end-to-end autonomous driving is an important and timely research direction.

[S2] The paper is well written and easy to follow.

[S3] The experiments and ablations are extensive and generally support the claims made in the paper.

[W1] The experiments that evaluate domain generalization rely on open-loop nuScenes metrics. While useful, open-loop evaluation is limited for assessing the robustness of end-to-end driving systems, particularly for planning components where closed-loop feedback effects play a crucial role. Additional closed-loop evaluations would strengthen the empirical evidence.

[W2] The impact of RoCA on training and inference time could be expanded. See [Q1] and [Q2].

[W3] The related work section could be expanded. Connections to causal learning and other approaches addressing domain generalization could be discussed. You mentioned VLP is one of the closest approaches to yours, but it’s discussed only in the appendix.

---

> ### Author Rebuttal · Authors · 2026-03-30
>
> We thank the reviewer for the constructive feedback on **closed‑loop domain generalization**, **runtime efficiency**, and **baseline selection**, as well as for highlighting the importance of **evaluating robustness under realistic, interactive driving conditions**. These comments helped us strengthen both the empirical evaluation and the clarity of the presented results.
>
> **W1: Domain generalization closed-loop evaluation**
>
> **Response:** We agree that **open‑loop nuScenes metrics alone are insufficient** to assess real‑world robustness for cross‑domain end‑to‑end driving. To address this, we add **closed‑loop cross‑domain evaluations** by training on **Bench2Drive (source)** and evaluating on **NAVSIM v1 and DriveArena (targets)** using **planning‑oriented closed‑loop metrics**—**PDMS↑**, **Ego‑Progress (EP↑)**, **No‑at‑fault Collisions (NC↑)**, and **Route Completion (RC↑)**. As shown in **Tables R4/R5**, RoCA consistently improves **planning quality, ego progress, and safety** over the corresponding base planners, demonstrating **stronger robustness under closed‑loop feedback and domain shift**.
>
> **W2 and Q1: Training and inference time impact**
>
> **Response:** **RoCA adds no inference overhead by default**: the **Gaussian Process is used only during training/fine‑tuning**, and the deployed planner runs unchanged (**no GP operations, no covariance inversions, no added latency**). For completeness, we also report an **optional GP‑at‑inference mode** with modest overhead since it operates on **small, fixed‑size matrices**; latency is reported in **Appendix Table 7 p. 14**. This mode is **strictly optional and not required for deployment**.
> We further analyze the impact of **codebook size** in **Appendix Table 10 (p.15)** and report training and optional inference time when varying **Nego** (with **fixed Nagent = 64, C = 64**) in **Table R3**. Reducing **Nego** slightly lowers optional inference cost and has only a minor effect on training time, confirming that **RoCA’s runtime is stable and tunable** with respect to codebook size.
>
> Table R3. Training and inference (optional) time (varying Nego​; fixed Nagent=64, C=64).
> ```
> +----------------+-----------------------+-----------------------+
> |                | Nego =48, Nagent =64  | Nego =18, Nagent =64  |
> +----------------+-----------+-----------+-----------+-----------+
> | Method         | Training  | Inference | Training  | Inference |
> |                |           | Optional  | Training  | Optional  |
> +----------------+-----------+-----------+-----------+-----------+
> | SparseDrive    | 33 hrs    | 133 ms    | 33 hrs    | 133 ms    |
> | RoCA           | 38 hrs    | 167 ms*   | 37.5 hrs  | 160 ms*   |
> +----------------+-----------+-----------+-----------+-----------+
> ```
> *Inference shown is the optional GP‑at‑inference mode; default deployment matches the base planner’s latency.
>
>
> **W3: Related work discussion**
>
> **Response:** Thank you for the suggestion. We will expand the **related work section** to include stronger connections to **causal learning** and other **domain generalization approaches**, and move the discussion of closely related methods (e.g., **VLP**) into the main paper.
>
> **Q2: Baseline variants**
>
> **Response:** We selected **VAD‑Tiny** and **SparseDrive‑S** primarily for computational efficiency, enabling rapid iteration and extensive ablations within the submission timeline. To address potential selection bias, we additionally include **larger‑scale results with ORION**. Across all settings, RoCA shows **consistent performance gains**, indicating that improvements are **not specific to model size** and generalize from compact to larger E2E planners.
>
> **Q3: Active learning comparison limited to random sampling**
>
> **Response:** We agree that comparison against random sampling alone is insufficient. We therefore added **stronger active‑learning baselines**, including **MC Dropout** and other **uncertainty‑based sampling strategies**, evaluated under the same closed‑loop protocol. As shown in **Table R2 (Reviewer z7Yf)**, **RoCA’s GP‑based selection consistently outperforms these baselines**, demonstrating that gains arise from RoCA’s **GP‑over‑planner‑tokens design**, rather than uncertainty‑based sampling alone.

---

> > ### Author Rebuttal · Reviewer_HAAb · 2026-04-01
> >
> > Thanks for the detailed responses and additional experiments. My concerns are mostly addressed.
> >
> > I will raise the overall recommendation from "Weak Accept" to "Accept".

---

> > > ### Author Response · Authors · 2026-04-01
> > >
> > > We sincerely thank you for revisiting our submission, carefully considering the rebuttal and additional experiments, and for raising your overall recommendation. We greatly appreciate your thoughtful feedback and engagement, which helped us improve the clarity and strength of the paper.

---

### Official Review · Reviewer_z7Yf · 2026-03-12

**Soundness:** 3
**Presentation:** 3
**Significance:** 3
**Originality:** 3
**Overall Recommendation:** 4
**Confidence:** 3

**Summary:**

This paper proposes a cross-domain end-to-end autonomous driving framework called RoCA, whose core idea is to introduce Gaussian processes into trajectory prediction. By leveraging a set of learnable basis tokens and a trajectory codebook, it models the diversity of driving scenarios and provides uncertainty estimates during prediction. RoCA can serve as a regularization term during source domain training to improve model robustness, and it supports supervised/unsupervised adaptation as well as active learning in the target domain.

**Compliance With Llm Reviewing Policy:**

Affirmed.

**Final Justification:**

Most of my concerns have been satisfactorily addressed. Therefore, I will maintain my overall positive score.

**Key Questions For Authors:**

- Why Gaussian Process? Compared to other uncertainty modeling methods (e.g., Bayesian Neural Networks, Deep Ensembles), what are the advantages of Gaussian processes in cross-domain trajectory prediction? Are there theoretical comparisons or justifications to support this?

- How to scale GP to large-scale datasets? GPs require computing the inverse of the covariance matrix during inference, which has high computational complexity. How can efficiency be ensured for real-time deployment of AD?

- Is the initialization of the codebook critical? Is the performance sensitive to the initialization of the basis tokens and trajectory codebook? If the clustering quality is low, will it affect the prediction performance of the Gaussian process?

- Besides weather and urban changes, has the study considered other types of domain shifts, such as changes in sensor types or differences in driving behaviors?

**Limitations:**

See weakness.

**Strengths And Weaknesses:**

Strengths

- Introduces Gaussian processes into cross-domain trajectory prediction for end-to-end autonomous driving, offering a new perspective on uncertainty modeling.

- Supports multiple adaptation strategies (supervised/unsupervised/active learning), making it adaptable to various real-world deployment scenarios.

- Can be integrated into existing mainstream end-to-end models to enhance their cross-domain robustness.

Weaknesses

- The three-stage training process is cumbersome and time-consuming (e.g., requiring 48+6+20 epochs as mentioned in the paper), which hinders rapid iteration.

- The method heavily relies on the quality of the base model. The training of the RoCA module depends on tokens extracted by the base model. If the base model performs poorly, RoCA may struggle to be effective.

- The paper only uses the RBF kernel and does not systematically analyze the impact of different kernel choices or their performance in cross-domain tasks.

- The Gaussian process requires computing the inverse of the covariance matrix during inference, which may become a bottleneck in highly real-time scenarios, significantly limiting scalability.

- The proposed active learning strategy is relatively simplistic, relying solely on variance from the Gaussian process for sampling, without comparisons to other uncertainty estimation methods (e.g., MC Dropout, Ensemble).

---

> ### Author Rebuttal · Authors · 2026-03-30
>
> We thank the reviewer for the constructive feedback on **training efficiency**, **model dependence**, **GP design choices**, and **active learning**, which helped clarify RoCA’s scalability and robustness.
>
> **W1: Multi-stage training pipeline**
>
> **A1:** Multi‑stage training is standard in E2E autonomous driving systems such as **VAD, UniAD, ORION, and SparseDrive**, which rely on source‑domain pretraining followed by fine‑tuning. RoCA adds only **one lightweight Gaussian Process stage learn the codebook basis**, trained for **6 epochs (~3 hours)**, incurring minimal overhead. When using **public pretrained models**, the initial pretraining stage can be skipped, making RoCA’s training cost comparable to existing pipelines while improving **cross‑domain robustness**.
>
> **W2: Dependence on base model quality**
>
> **A2:** RoCA does not critically depend on base model quality, and the **codebook need not be learned from the same backbone**. As shown in **Table R1**, a weaker model (**VAD‑Tiny**) using a codebook learned from a stronger model (**ORION or SSR**) achieves substantial gains over using its own codebook (**Avg L2: 0.77→0.66; Avg Col: 0.20→0.13**). This demonstrates cross‑model generalization of the learned basis. RoCA consistently improves **VAD, SSR, ORION, and SparseDrive**, indicating gains stem from the **GP‑over‑planner‑tokens formulation**, not a specific architecture.
>
> Table R1. Unsupervised fine‑tuning on nuScenes showing cross‑model codebook generalization.
> ```
> | Method                                        | Avg L2 | Avg Col |
> +-----------------------------------------------+--------+---------+
> | RoCA (VAD-T) w/ codebook from RoCA (ORION)    | 0.67   | 0.14    |
> | RoCA (VAD-T) w/ codebook from RoCA (SSR)      | 0.73   | 0.16    |
> | RoCA (VAD-T) w/ codebook from RoCA (VAD-T)    | 0.77   | 0.20    |
> ```
>
> **W3: GP kernel choices**
>
> **A3:** We analyze kernel choices in **Appendix Table 9 (p.15)**, comparing **Linear, RBF, and Rational Quadratic** kernels. RoCA achieves **consistent performance across kernels**, indicating results are **not sensitive to kernel design** nor specific to RBF.
>
> **W4: GP inference cost**
>
> **A4:** RoCA requires **no GP computation at deployment**. The GP is used **only during training/fine‑tuning**; the deployed planner runs unchanged (**no covariance inversion, no added latency**). We additionally evaluate an **optional GP‑at‑inference mode** (Appendix Table 6 p. 12), where the anchor‑based GP inverts only **small matrices (Nego × Nego, C × C)** with **Nego=48, C=64** (Sec. 3.2.3, Sec. 4.1). Latency is reported in **Appendix Table 7 (p.14)**; this mode is **optional**.
>
> **W5: Active learning strategy**
>
> **A5:** We evaluate stronger baselines, including **MC Dropout** and **Deep Ensembles**, for sample selection. Across transfer directions and labeling budgets, **RoCA + RoCA sampling achieves the lowest Avg. L2**, demonstrating the advantage of **GP‑based uncertainty selection**.
>
> Table R2. Active learning comparison with uncertainty estimation methods for sampling using Avg. L2 (m) metrics, Here S means Singapore and B means Boston.
>
> ```
> | Adapt Method     | Sampling    | S->B (10%) | S->B (15%) | B->S (10%) | B->S (15%) |
> | Direct finetune  | MC Dropout  | 0.729      | 0.694      | 0.814      | 0.806      |
> | Direct finetune  | Ensemble[A] | 0.708      | 0.683      | 0.802      | 0.781      |
> | RoCA             | Ensemble[A] | 0.577      | 0.539      | 0.641      | 0.608      |
> | RoCA             | RoCA        | 0.554      | 0.513      | 0.604      | 0.561      |
> ```
> [A] Simple and Scalable Predictive Uncertainty Estimation using Deep Ensembles, Lakshminarayanan et al., NeuRIPS, 2017.
>
> **Q1: Why Gaussian Processes?**
>
> **A1:** We use a **Gaussian Process** for its **closed‑form mean and variance**, yielding smooth pseudo‑labels and **well‑calibrated uncertainty** in low‑label, cross‑domain settings. Unlike **MC Dropout** or **Deep Ensembles**, a GP provides **one‑shot uncertainty** and supports **kernel design for domain transfer**, enabling **data‑efficient adaptation**.
>
> **Q2: Scalability to large datasets**
>
> **A2:** RoCA scales to large target datasets via **basis codebook fine‑tuning**. Once learned on the source domain, **unsupervised fine‑tuning adapts basis for target domain via RoCA**, enabling scalable adaptation to large datasets.
>
>
> **Q3:  Codebook initialization Sensitivity**
>
> **A3:** In **Appendix Table 10**, varying the number of basis tokens (**3×6 vs. 3×16**) shows only modest performance differences, indicating **robustness to codebook initialization**.
>
>
> **Q4: Other domain shifts**
>
> **A4:** We evaluate **simulation‑to‑real transfer** from **Bench2Drive to nuScenes** (Table 2) and extend this to **Bench2Drive → NAVSIM / DriveArena** (see **Tables R4 and R5** in the response to Reviewer 5nER).

---

> > ### Author Rebuttal · Reviewer_z7Yf · 2026-04-01
> >
> > Thank you to the authors for the detailed response. Most of my concerns have been satisfactorily addressed. Therefore, I will maintain my overall positive score.

---

> > > ### Author Response · Authors · 2026-04-01
> > >
> > > We sincerely thank you for your careful consideration of our rebuttal and for confirming that the raised concerns have been resolved. We appreciate your constructive engagement and overall positive evaluation.

---

### Decision · Program_Chairs · 2026-04-30

**Decision:**

Accept (regular)

**Comment:**

RoCA presents a Gaussian Process module on learned token/trajectory codebooks for cross-domain end-to-end self-driving, providing probabilistic prediction, uncertainty estimation, and plug-and-play adaptation without extra inference cost. All reviewers praise the principled GP formulation, versatility across backbones (VAD, SparseDrive, ORION), and strong cross-domain results. Initial concerns about missing closed-loop evaluation were addressed with NAVSIM and DriveArena experiments; comparisons against contrastive learning and BNN baselines were added. All four reviewers confirmed concerns resolved, with reviewer HAAb raising to Accept. The individual components are established, but their integration for joint prediction and domain adaptation is well-validated, which is a solid point in the ICML community.